# Transferable Reinforcement Learning via Probabilistic Latent Embeddings and Dynamic Policy Adaptation for Sim-to-Real Deployment

**Gengyue Han** [1 2]  **Yiheng Feng** [1]

## Abstract

Due to limited resources and public safety concerns, deep reinforcement learning (RL) agents for many cyber-physical systems (e.g., autonomous vehicles) are first trained in simulators. However, when deployed in real world environments, they often suffer from performance degradation or safety violations because of the inevitable *Sim2Real* gap. Existing zero-shot approaches, such as robust safe RL and domain randomization, mitigate this issue but typically at the cost of degraded performance or residual safety risks when experiencing unmodeled system dynamics. To address these limitations, we propose a novel reinforcement learning framework that enables safe and efficient policy transfer via probabilistic latent embeddings and dynamic policy adaptation. We consider a family of Constrained Markov Decision Processes (CMDPs) under different environment contexts. By leveraging latent context variable in meta-RL, the proposed framework infers the latent representation of the environment from simulated experiences. Furthermore, it incorporates a distributional RL formulation, which allows risk levels of the deployed policy to be adjusted dynamically, based on the estimation accuracy of the latent context variable. This strategy promotes safety at the early deployment stage and improves efficiency through fast policy adaptation under the *Sim2Real* gap.

## 1. Introduction

Reinforcement learning (RL) has emerged as a promising paradigm for decision-making in cyber-physical systems, such as robotics, healthcare, and autonomous driving (Tang et al., 2025; Jayaraman et al., 2024; Han et al., 2026). However, RL training typically requires substantial interactions between the agent and the environment, making direct training on real systems expensive and often unsafe due to exploratory behaviors. As a result, many RL policies are first trained in simulators, where data collection is cheap and safety risks are minimized (Ray et al., 2019). However, when deployed in the real world, such policies trained in simulation often suffer from performance degradation and safety risks due to the mismatch between simulation and reality. This mismatch is commonly referred to as the *Sim2Real* gap (Da et al., 2025; Aljalbout et al., 2025).

Existing works try to address the *Sim2Real* challenge from multiple perspectives. Domain randomization strategies improve robustness by exposing policies to diverse simulated conditions, but they do not provide explicit safety guarantees and may lead to unstable training when the simulated variability is overly aggressive (Tobin et al., 2017). Distributionally robust and risk-sensitive RL methods formulate *Sim2Real* transfer as a worst-case optimization problem, enhancing reliability under severe distribution shifts, but at the cost of overly conservative policies and degraded performance (As et al., 2025; Pinto et al., 2017). Domain adaptation and meta-RL approaches seek to reduce the *Sim2Real* gap by aligning representations or inferring latent environment contexts, yet most existing methods assume fixed risk preferences and lack mechanisms to regulate safety–performance trade-offs at the deployment stage (Rakelly et al., 2019; Yu et al., 2020). In summary, current approaches either sacrifice efficiency for robustness or fail to ensure safety when encountering unmodeled real-world dynamics.

To overcome these limitations, we propose a unified framework for safe and adaptive *Sim2Real* policy transfer. We formulate a family of constrained Markov decision processes (CMDPs) under varying environment contexts and dynamically adjust the risk level of the deployed policy to balance safety and performance during real-world deployment.

Specifically, the key innovations are summarized as follows:

- Unified safe and adaptive *Sim2Real* framework: We

---

[1]Lyles School of Civil and Construction Engineering, Purdue University, West Lafayette, USA [2]Elmore Family School of Electrical and Computer Engineering, Purdue University, West Lafayette, USA. Correspondence to: Yiheng Feng <feng333@purdue.edu>.

*Proceedings of the 43rd International Conference on Machine Learning*, Seoul, South Korea. PMLR 306, 2026. Copyright 2026 by the author(s).

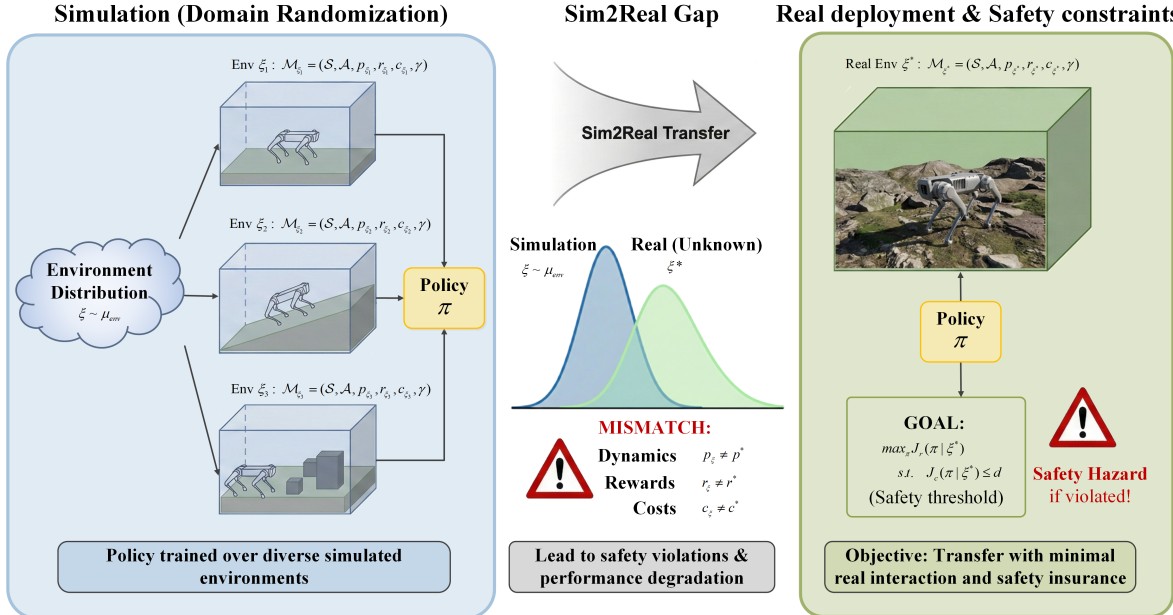

*Figure 1.* The *Sim2Real* problem statement of RL policy

present a unified CMDP-based framework that integrates probabilistic latent context variable adaptation with distributional reinforcement learning, enabling safe and adaptive policy transfer under *Sim2Real* mismatch.

- Inference time risk regulation: By incorporating distributional RL into the CMDP formulation, our approach allows the risk levels of the deployed policy to be adjusted dynamically at inference time, balancing safety and performance during real-world deployment.

- Theoretical guarantees: We provide theoretical proofs on training convergence, quantify the benefits of latent context variable adaptation, and demonstrate assured safety under the *Sim2Real* gap.

## 2. Related Work

**Safe RL.** Safe RL is commonly formulated as a CMDP, where the objective is to maximize expected return while satisfying cumulative cost constraints (Altman, 2021; Achiam et al., 2017; Wachi et al., 2024; Kushwaha et al., 2025). Existing CMDP methods broadly fall into Lagrangian-based approaches, which balance reward and cost via dual variables (Achiam et al., 2017; Hong et al., 2024), and constraint-satisfying approaches that enforce feasibility during policy learning (Mestres et al., 2025; Wachi, 2021). Beyond CMDPs, alternative paradigms such as risk-sensitive (Prashanth & Michael, 2022; Wang et al., 2024) and robust RL (Jaimungal et al., 2022; Meggendorfer et al., 2025)

have also been explored, though they often provide limited flexibility in balancing safety and performance during deployment. Within the CMDP framework, distributional RL (Bellemare et al., 2017; Dabney et al., 2018b) enables explicit risk level control while preserving safety constraints (Yang et al., 2023), which motivates this work.

***Sim2Real* transfer.** A variety of approaches have been proposed to address *Sim2Real* transfer in RL (Da et al., 2025). Some works focus on domain randomization and curriculum-based strategies, which expose policies to diverse simulated conditions to improve robustness, but lack explicit safety guarantees and may lead to unstable training under aggressive randomization (Chen et al., 2021; Tobin et al., 2017; Danesh et al., 2025). Another prominent line of work formulates *Sim2Real* transfer as a distributionally robust learning problem, optimizing policies against worst-case transition dynamics within bounded uncertainty sets, often combined with pessimistic evaluation or ensemble-based uncertainty estimation (Liu et al., 2024; Liu & Xu, 2024a;b; As et al., 2025). While these methods improve reliability under severe distribution shifts, their emphasis on worst-case scenarios can result in overly conservative policies and degraded task performance. To overcome these limitations, domain adaptation methods explicitly align simulated and real environments at the representation or distributional level (Farhadi et al., 2024), reducing mismatch through feature alignment (Tzeng et al., 2017; Sodhani et al., 2021), latent state or dynamics learning (Rakelly et al., 2019; Yu et al., 2020; Guo et al., 2025), representation-mismatch

regularization (Lyu et al., 2024), and adversarial objectives (Bousmalis et al., 2018; Ganin et al., 2016).

**Meta-RL.** Meta-learning aims to acquire a learning mechanism that rapidly adapts to new tasks using only a small amount of data (Schmidhuber, 1992). Meta-RL is a specialization of meta-learning in reinforcement learning, where the goal is to quickly adapt to new MDPs or environment dynamics (Finn et al., 2017; Duan et al., 2016; Wang & Sun, 2022; Xu & Zhu, 2026). Prevalent meta-RL methods are categorized into three groups: gradient-based methods, which perform fast adaptation through explicit inner-loop parameter updates (e.g., MAML-TRPO (Finn et al., 2017)); memory-based methods, which rely on recurrent architectures to implicitly encode task information in their internal states (e.g., RL$^2$ (Duan et al., 2016)); and latent context variable-based methods, which introduce a task-dependent latent context inferred from experience to condition the policy (e.g., PEARL (Rakelly et al., 2019), VariBAD (Zintgraf et al., 2021)). Within the latent context variable-based category, a prominent line of work adopts a Bayesian perspective, where adaptation is achieved via probabilistic inference over latent task variables rather than online policy optimization (Cho et al., 2024; de Vries et al., 2025). They model task uncertainty through posterior inference, enabling structured exploration and rapid adaptation under partial observability.

## 3. Problem Statement and Solution Framework

We consider a family of CMDPs indexed by an environment parameter $\xi \in \Xi$. Each $\xi$ specifies unique transition dynamics, reward and cost returns of an environment:

$$\mathcal{M}_\xi = (\mathcal{S}, \mathcal{A}, p_\xi, r_\xi, c_\xi, \gamma), \tag{1}$$

where $\mathcal{S}$ and $\mathcal{A}$ are state and action spaces. In the environment with $\xi$, $p_\xi$ is the state transition function, $r_\xi : \mathcal{S} \times \mathcal{A} \rightarrow [0, r_{\xi,\max}]$ denotes the reward function, and $c_\xi : \mathcal{S} \times \mathcal{A} \rightarrow [0, c_{\xi,\max}]$ denotes the cost function. $\gamma \in [0, 1)$ is the discount factor. As shown in Figure 1, in simulation, environments with $\xi \sim \mu_{env}$ are sampled through domain randomization. At deployment, the agent operates in a fixed real environment $\xi^*$, which is unknown in the training process. With control policy $\pi$ obtained from environment $\mathcal{M}_{\xi^*}$, the expected discounted reward is $J_r(\pi|\xi) = \mathbb{E}[\sum_{t=0}^\infty \gamma^t r_\xi(\boldsymbol{s}, \boldsymbol{a})]$, and the expected discounted cost is $J_c(\pi|\xi) = \mathbb{E}[\sum_{t=0}^\infty \gamma^t c_\xi(\boldsymbol{s}, \boldsymbol{a})]$. Our goal to close the *Sim2Real* gap is equivalent to train an optimal policy $\pi$ that satisfies,

$$max_\pi J_r(\pi|\xi^*) \quad s.t. \quad J_c(\pi|\xi^*) \leq d, \tag{2}$$

where $d$ is a cost threshold. Notably, this optimal policy is trained in simulations with $\xi \sim \mu_{env}$, which is different

from the real $\xi^*$. This mismatch leads to potential safety hazard in the real environment $\mathcal{M}_{\xi^*}$. In addition, the cost of online training at the deployment stage is high and only a limited number of experiences can be collected with a reasonable budget. To address these two challenges, this work aims to develop a transferable reinforcement learning agent that is trained entirely in simulation while remaining adaptive and safe at deployment. An overview of the *Sim2Real* transfer framework is summarized in Figure 2. We introduce an encoder that extracts salient environment-specific information, enabling the learned policy to condition its actions on the underlying environment (Section 4.1). In addition, distributional reinforcement learning is employed to characterize the distributions of both rewards and costs under the latent information (Section 4.2 and 4.3), allowing risk level to be adjusted with latent context variable adaptation (Section 5.1). A safety upper-bound is then developed (Section 5.2) and the agent is proven to be safe under dynamic adaptation of risk-sensitive policy (Section 5.3 and 5.4). Together, these components enables safe and effective policy transfer with limited real-world interactions under *Sim2Real* gap.

## 4. Environment Encoder and RL Agent Design

Formal proofs of the theorems, propositions, and lemmas in this section are provided in Appendix A.

### 4.1. Latent Context Design

A latent context variable $\boldsymbol{z}$ is introduced to encode salient information of the environment $\mathcal{M}_{\xi^*}$. Inspired by (Rakelly et al., 2019), an inference network encoder $q_\phi(\boldsymbol{z}|\boldsymbol{D}_\xi)$ is trained to estimate $\boldsymbol{z}$, where $\boldsymbol{D}_\xi = \{(\boldsymbol{s}_i, \boldsymbol{a}_i, \boldsymbol{s}'_i, r_i, c_i)\}_{i=1}^N$ is a context set with $N$ transition samples. The encoder can be optimized in a model-free manner to approximate the state-action value functions in CMDPs. Both the reward and cost likelihood are considered in the variational lower bound. Specifically, with collected context set $\boldsymbol{D}_\xi$, a permutation-invariant representation for $q_\phi(\boldsymbol{z}|\boldsymbol{D}_\xi)$ is modeled as a product of independent factors,

$$q_\phi(\boldsymbol{z}|\boldsymbol{D}_\xi) \propto \prod_{i=1}^N \Psi_\phi(\boldsymbol{z}|\boldsymbol{D}_{\xi,i}), \tag{3}$$

where $\boldsymbol{D}_{\xi,i}$ represents $i$-th transition in the context set $\boldsymbol{D}_\xi$. $\Psi_\phi(\boldsymbol{z}|\boldsymbol{D}_{\xi,i}) = \mathcal{N}(f_\phi^{\boldsymbol{\mu}}(\boldsymbol{D}_{\xi,i}), f_\phi^{\boldsymbol{\sigma}}(\boldsymbol{D}_{\xi,i}))$ is a Gaussian factor. The function $f_\phi$ is a neural network parameterized by $\phi$, which estimate the mean vector $\boldsymbol{\mu}$ and diagonal variance $\boldsymbol{\sigma}$ for $\boldsymbol{D}_{\xi,i}$. Notably, the product of independent Gaussian factors $\Psi_\phi(\boldsymbol{z}|\boldsymbol{D}_{\xi,i})$ also result in a Gaussian distribution (i.e., the encoder $q_\phi(\boldsymbol{z}|\boldsymbol{D}_\xi)$). The optimization of $\phi$ depends on the gradients of both the reward and cost critics, and an

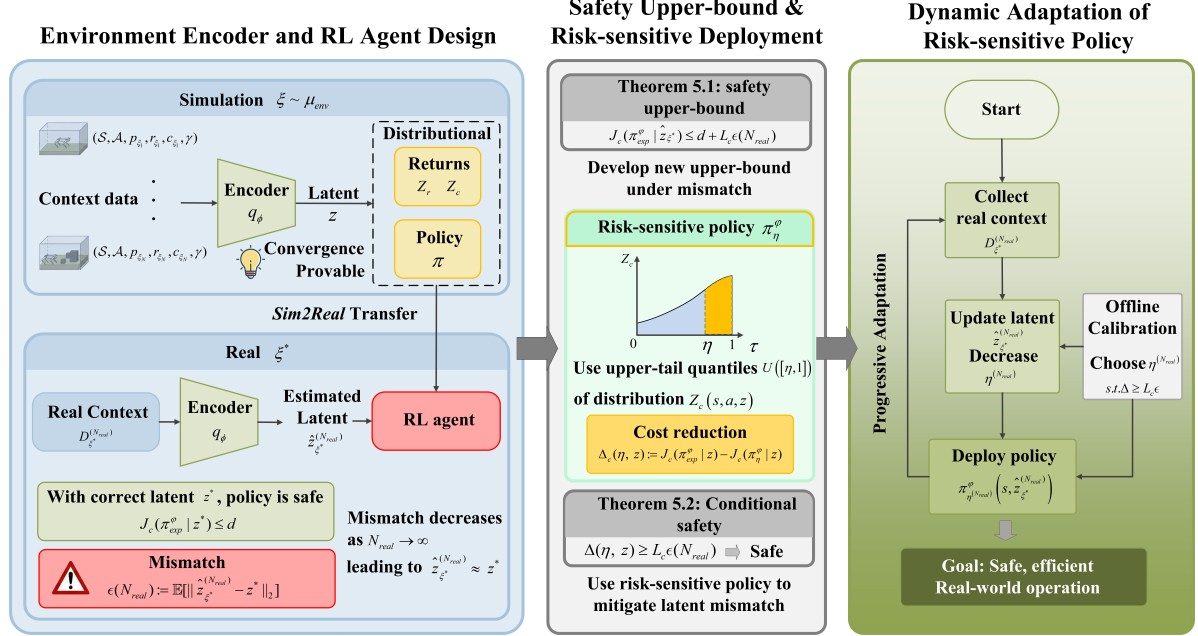

*Figure 2.* Framework of *Sim2Real* transfer via latent context variable adaptation

information bottleneck as shown below,

$$\mathcal{L}^{\text{encoder}}(\phi) = \beta_r \mathcal{L}_r^{\text{critic}} + \beta_c \mathcal{L}_c^{\text{critic}} + \beta_{KL} \mathcal{L}_{KL}, \quad (4)$$

where $\beta_r$, $\beta_c$, and $\beta_{KL}$ are constant parameters. $\mathcal{L}_r^{\text{critic}}$ and $\mathcal{L}_c^{\text{critic}}$ are reward and cost critic losses, respectively. $\mathcal{L}_{KL}$ is the information bottleneck derived by KL-divergence between the encoder and a Gaussian prior $p(\boldsymbol{z}) = \mathcal{N}(\mathbf{0}_{d_z}, I_{d_z})$:

$$\mathcal{L}_{KL} = \text{KL}(q_\phi(\boldsymbol{z}|\boldsymbol{D}_\xi)||p(\boldsymbol{z})), \quad (5)$$

where $d_z$ is the dimension of the latent context. The bottleneck constrains $\boldsymbol{z}$ to contain only information from the context that is necessary to adapt to the task at hand, mitigating overfitting to training tasks.

By minimizing $\mathcal{L}^{\text{encoder}}(\phi)$, the encoder $q_\phi(\boldsymbol{z}|\boldsymbol{D}_\xi)$ could accurately estimate the latent context variable $\boldsymbol{z}$ given a collected context set $\boldsymbol{D}_\xi$, which leads to $J_r(\pi|\xi) \approx J_r(\pi|\boldsymbol{z})$ and $J_c(\pi|\xi) \approx J_c(\pi|\boldsymbol{z})$. For an environment with $\xi$, the latent context estimation with $N$ transition samples is,

$$\hat{\boldsymbol{z}}_\xi^{(N)} = \mathbb{E}_{q_\phi(\boldsymbol{z}|\boldsymbol{D}_\xi^{(N)})}[\boldsymbol{z}] \in \mathbb{R}^{d_z}. \quad (6)$$

### 4.2. Distributional RL Formulation

In this work, implicit quantile network (IQN) (Dabney et al., 2018a) is applied to optimize the CMDP, from which the distribution over returns could be modeled. Let $F_Z^{-1}(\tau)$, $\tau \in [0, 1]$ be the quantile function for the random variable $Z$, i.e. $Z_\tau := F_Z^{-1}(\tau)$. The expectation of $Z_\tau$ is given by $Q(\boldsymbol{s}, \boldsymbol{a}) := \mathbb{E}_{\tau \sim U([0,1])}[Z(\boldsymbol{s}, \boldsymbol{a}; \tau)]$. Sample $\tau_i$ and $\tau_j$ from two independent distributions, $\tau_i, \tau_j \sim U([0, 1])$. The quantile TD-errors for the reward and cost functions are derived as follows,

$$\delta_{ij}^r = r + \gamma Z_{\tau_j}^{\theta_r^-}(\boldsymbol{s}', \pi^\varphi(\boldsymbol{s}', \boldsymbol{z}), \boldsymbol{z}) - Z_{\tau_i}^{\theta_r}(\boldsymbol{s}, a, \boldsymbol{z}), \quad (7)$$

$$\delta_{ij}^c = c + \gamma Z_{\tau_j}^{\theta_c^-}(\boldsymbol{s}', \pi^\varphi(\boldsymbol{s}', \boldsymbol{z}), \boldsymbol{z}) - Z_{\tau_i}^{\theta_c}(\boldsymbol{s}, a, \boldsymbol{z}). \quad (8)$$

Then the Huber quantile regression loss (Huber, 1992) can be calculated as:

$$\mathcal{L}_r^{\text{critic}}(\theta_r) = \frac{1}{NN'} \sum_{i=1}^{N} \sum_{j=1}^{N'} \rho_{\tau_i}(\delta_{ij}^r), \quad (9)$$

$$\mathcal{L}_c^{\text{critic}}(\theta_c) = \frac{1}{NN'} \sum_{i=1}^{N} \sum_{j=1}^{N'} \rho_{\tau_i}(\delta_{ij}^c), \quad (10)$$

with

$$\rho_{\tau_i}(\delta_{ij}) = |\tau_i - \mathbb{I}(\delta_{ij} < 0)| \cdot \frac{l(\delta_{ij})}{\kappa}, \quad (11)$$

$$l(\delta_{ij}) = \begin{cases} \frac{1}{2}\delta_{ij}^2, & \text{if} \quad |\delta_{ij}| < \kappa, \\ \kappa(|\delta_{ij}| - \frac{1}{2}\kappa), & \text{else}. \end{cases} \quad (12)$$

where $N$ and $N'$ are number of i.i.d. samples of $\tau_i$ and $\tau_j$, respectively. $\kappa$ is a threshold that controls the transition point between L2 behavior for small TD errors and L1

behavior for large TD errors, improving both stability and robustness in quantile regression.

To enable continuous action making, a deterministic actor $\pi^{\varphi}(s, z)$ is trained with a risk-neutral objective based on the expectation of IQN critics over $\tau \sim U([0, 1])$. Specifically, sampling $\tau_r$ and $\tau_c$ randomly for $K$ times from $U([0, 1])$, the actor loss is,

$$
\begin{aligned}
\mathcal{L}^{\text{actor}}(\varphi) = -\frac{1}{K}\mathbb{E}_{(s,z)}[\sum_{k=1}^{K} Z_{\tau_r}^{\theta_r}(s, \pi^{\varphi}(s, z), z) \\
- \lambda_L \sum_{k=1}^{K} Z_{\tau_c}^{\theta_c}(s, \pi^{\varphi}(s, z), z)],
\end{aligned}
\tag{13}
$$

where $\lambda_L$ is a Lagrangian multiplier (Stooke et al., 2020). Since $\tau \sim U([0, 1])$, the learned policy is risk-neutral, which could be denoted as $\pi_{exp}^{\varphi}(s, z)$.

### 4.3. Convergence and Adaptation with Latent Context

#### 4.3.1. TRAINING CONVERGENCE

We prove that conditioning on $z$ preserves the contraction property of distributional Bellman operators, hence preserving the standard convergence property in IQN.

Define the distributional Bellman operator acting on return distributions as:

$$
(\mathcal{T}_r^{\pi,z} Z)(s, a) \overset{D}{=} r(s, a, z) + \gamma Z(s', \pi(s', z), z), \tag{14}
$$

and analogously define $\mathcal{T}_c^{\pi,z}$ for costs by replacing $r$ with $c$. Let $W_p(\cdot, \cdot)$ denote the $p$-Wasserstein distance on distributions.

**Lemma 4.1** (Latent-conditioned contraction). *Fix any $z$ and any policy $\pi(\cdot, z)$. Define $\bar{d}_p(Z_1, Z_2) := \sup_{s,a} W_p(Z_1(s, a, z), Z_2(s, a, z))$. Then the latent-conditioned distributional Bellman operator is a $\gamma$-contraction:*

$$
\bar{d}_p(\mathcal{T}_r^{\pi,z} Z_1, \; \mathcal{T}_r^{\pi,z} Z_2) \leq \gamma \, \bar{d}_p(Z_1, Z_2), \tag{15}
$$

*and the same holds for $\mathcal{T}_c^{\pi,z}$.*

**Theorem 4.2** (Adding latent context $z$ preserves Bellman contraction and fixed-point well-posedness). *For any fixed latent $z$ and any fixed policy $\pi(\cdot, z)$, the latent-conditioned distributional Bellman operators $\mathcal{T}_r^{\pi,z}$ and $\mathcal{T}_c^{\pi,z}$ are $\gamma$-contractions under $\bar{d}_p$. Consequently, each operator admits a unique fixed point $Z_r^{\pi,z}$ and $Z_c^{\pi,z}$ satisfying*

$$
Z_r^{\pi,z} = \mathcal{T}_r^{\pi,z} Z_r^{\pi,z}, \qquad Z_c^{\pi,z} = \mathcal{T}_c^{\pi,z} Z_c^{\pi,z}. \tag{16}
$$

*Moreover, for any initial return distributions $Z_{r,0}(\cdot, \cdot, z)$ and $Z_{c,0}(\cdot, \cdot, z)$, the iterates*

$$
Z_{r,k+1} := \mathcal{T}_r^{\pi,z} Z_{r,k}, \qquad Z_{c,k+1} := \mathcal{T}_c^{\pi,z} Z_{c,k}, \tag{17}
$$

*converge geometrically:*

$$
\begin{aligned}
\bar{d}_p(Z_{r,k}, Z_r^{\pi,z}) \leq \gamma^k \, \bar{d}_p(Z_{r,0}, Z_r^{\pi,z}), \\
\bar{d}_p(Z_{c,k}, Z_c^{\pi,z}) \leq \gamma^k \, \bar{d}_p(Z_{c,0}, Z_c^{\pi,z}).
\end{aligned}
\tag{18}
$$

The contraction in Theorem 4.2 characterizes the target for the IQN critics when $z$ is fixed. In practice, $z$ is generated by the encoder $q_\phi(z|D_\xi)$, and $\phi$ is trained jointly with the critics. We adopt an alternating frozen scheme: when updating the encoder, we freeze the critic parameters, and when updating the critics, the encoder is fixed. Specifically, at iteration $k$, we perform one critic step with $\phi_k$ fixed:

$$
\begin{aligned}
\theta_{r,k+1} = \theta_{r,k} - \alpha_k \widehat{\nabla}_{\theta_r} \mathcal{L}_r^{\text{critic}}(\theta_{r,k}; \phi_k), \\
\theta_{c,k+1} = \theta_{c,k} - \alpha_k \widehat{\nabla}_{\theta_c} \mathcal{L}_c^{\text{critic}}(\theta_{c,k}; \phi_k).
\end{aligned}
\tag{19}
$$

followed by one encoder step with critics frozen (i.e., $\theta_{r,k+1}, \theta_{c,k+1}$ fixed):

$$
\phi_{k+1} = \phi_k - \beta_k \widehat{\nabla}_\phi \mathcal{L}^{\text{encoder}}(\phi_k; \theta_{r,k+1}, \theta_{c,k+1}), \tag{20}
$$

where $\mathcal{L}^{\text{encoder}}$ is defined in (4). This prevents direct interference on critic parameters and yields a stable block-coordinate stochastic approximation.

For analysis, define a Lyapunov function

$$
\begin{aligned}
\mathcal{J}(\theta_r, \theta_c, \phi) := \beta_r \mathcal{L}_r^{\text{critic}}(\theta_r; \phi) + \beta_c \mathcal{L}_c^{\text{critic}}(\theta_c; \phi) \\
+ \beta_{KL} \mathcal{L}_{KL}(\phi).
\end{aligned}
\tag{21}
$$

where $\mathcal{L}_r^{\text{critic}}$ and $\mathcal{L}_c^{\text{critic}}$ are given in (9)–(12), and the dependence on $\phi$ is through $z \sim q_\phi(z|D_\xi)$.

**Lemma 4.3** (Descent lemma (smooth upper bound)). *Assume $\mathcal{J}$ has $L$-Lipschitz gradients, i.e., $\|\nabla \mathcal{J}(u) - \nabla \mathcal{J}(v)\|_2 \leq L\|u - v\|_2$ for all $u, v$. Then for any $x$ and any $\Delta$,*

$$
\mathcal{J}(x + \Delta) \leq \mathcal{J}(x) + \langle \nabla \mathcal{J}(x), \Delta \rangle + \frac{L}{2}\|\Delta\|_2^2. \tag{22}
$$

Based on Lemma 4.3, the following Theorem that demonstrates the stability of the alternating frozen scheme for joint training is stated.

**Theorem 4.4** (Alternating frozen training is stable under constant step sizes). *Under the alternating frozen updating scheme with constant step sizes $\alpha_k \equiv \alpha > 0$ and $\beta_k \equiv \beta > 0$, let $g_k := \nabla \mathcal{J}(\theta_{r,k}, \theta_{c,k}, \phi_k)$. Then,*

$$
\limsup_{K \to \infty} \frac{1}{K} \sum_{k=0}^{K-1} \mathbb{E}[\|g_k\|_2^2] \leq O(\max\{\alpha, \beta\}). \tag{23}
$$

*Equivalently, the iterates approach an $O(\max\{\alpha, \beta\})$-stationary neighborhood of $\mathcal{J}$, and the joint training is stable (does not diverge).*

### 4.3.2. DEPLOYMENT ADAPTATION

In the last section, we show that introducing the latent context $z$ preserves training stability. Moreover, at deployment, conditioning the policy on $z$ resolves task aliasing and eliminates the irreducible performance gap of non-contextual policies.

Consider a policy that does not use context, i.e., $\pi(a|s)$. Even with unlimited data, such a policy cannot in general achieve per-task optimal behavior when different environments require different optimal actions in the same state (*task aliasing*).

**Proposition 4.5** (Irreducible regret under task aliasing without $z$). *Suppose there exist two environments $\xi_1, \xi_2 \in \text{supp}(\mu_{env})$ sharing a common state $\bar{s}$ and two actions $a_1, a_2$ such that*

$$
\begin{aligned}
Q_r^{\xi_1}(\bar{s}, a_1) - Q_r^{\xi_1}(\bar{s}, a_2) &\geq \delta, \\
Q_r^{\xi_2}(\bar{s}, a_2) - Q_r^{\xi_2}(\bar{s}, a_1) &\geq \delta
\end{aligned}
\tag{24}
$$

*for some $\delta > 0$. Then for any non-contextual policy $\pi(a|s)$, there exists a task $\xi \in \{\xi_1, \xi_2\}$ such that the (one-step) regret at $\bar{s}$ is at least $\delta/2$. In particular, this gap does not vanish with more data.*

*Remark* 4.6. Proposition 4.5 is stated in terms of reward regret. The same task-aliasing phenomenon also affects safety: when the constraint-satisfying actions differ across environments in the same state, a non-contextual policy $\pi(a|s)$ cannot simultaneously avoid unsafe actions in all tasks.

The proposition above identifies the limitation of non-contextual policies. In contrast, a contextual policy $\pi(a|s, z)$ can choose its action depending on the inferred environment context. Thus, when $z$ correctly captures the difference between $\xi_1$ and $\xi_2$, the policy will choose $a_1$ in $\xi_1$ and $a_2$ in $\xi_2$, removing the task-aliasing regret. In practice, however, if the estimated $z$ is inaccurate, the policy may still choose an action suited to the wrong environment and cause cost and safety violations. This motivates the next section, where $z$ is refined online for *Sim2Real* transfer.

## 5. *Sim2Real* Transfer via Latent Context Variable Adaptation

Formal statements and proofs of the definitions, assumptions, theorems, and lemmas in this section are provided in Appendix A.

### 5.1. Latent Context Variable Adaptation

The model mismatch between simulation and the real environment may cause safety threshold underestimation. The latent context design in Section 4.1 aims to excavate environmental information and mitigating this mismatch by training

a latent context variable $z$. Specifically, a well-trained policy is assumed to be safe in various simulation environments with estimated latent $\hat{z}$ (Assumption A.3). When the well-trained policy is transferred to the real environment with latent $z^*$, the performance should be satisfactory if the *actor* network takes actions under $z^*$, given the following assumption 5.1.

**Assumption 5.1** (Real environment close to training ensemble). The real environment $\xi^*$ does not necessarily to be the same as the environments in the training process. However, the gap between simulations and the real environment could not be too far. $\xi^*$ should falls in the domain randomization distribution $\mu_{env}$.

Under this assumption, $N_{real}$ context samples are collected online to refine $\hat{z}$ to $\hat{z}_{\xi^*}$. If $N_{real}$ is large enough, $\hat{z}_{\xi^*} \approx z^*$.

### 5.2. *Sim2Real* Safety Upper-bound

However, if the number $N_{real}$ is not sufficient, there is an inherent mismatch between $\hat{z}_{\xi^*}$ and $z^*$, which is defined as $\epsilon(N_{real}) := \mathbb{E}[||\hat{z}_{\xi^*}^{(N_{real})} - z^*||_2]$ (Definition A.1). The mismatch may lead to a cost threshold violation. However, this violation has an upper-bound as defined below.

**Theorem 5.2** (*Sim2Real* safety upper-bound under risk-neutral deployment). *With risk-neutral policy $\pi_{exp}^\varphi(s, z)$, the cost of the deployed policy satisfies,*

$$
J_c(\pi_{exp}^\varphi | \hat{z}_{\xi^*}) \leq d + L_c \epsilon(N_{real}),
\tag{25}
$$

*where $L_c$ is a Lipschitz constant of the cost value function. The item $L_c \epsilon(N_{real})$ characterizes the extra cost introduced by the mismatch between $\hat{z}_{\xi^*}$ and $z^*$. In particular, if the mismatch becomes negligible ($\epsilon(N_{real}) \approx 0$), then the deployed risk-neutral policy satisfies the cost threshold $d$ adopted in the training process.*

### 5.3. Cost Reduction via Risk-sensitive Deployment

To ensure sufficient safety margin at deployment, a risk-sensitive policy $\pi_\eta^\varphi$ that is more conservative than $\pi_{exp}^\varphi$ shall be used, where $\eta \in (0, 1)$ constrains the lower-bound of $\tau$. Instead of sampling $\tau$ from $U([0, 1])$, we evaluate the cost critic using upper-tail quantiles $U([\eta, 1])$, which emphasizes high-cost outcomes and yields more conservative deployment actions. In particular, the induced upper-tail cost value $Q_c^\eta(s, a, z) := \mathbb{E}_{\tau \sim U([\eta, 1])}[Z^{\theta_c}(s, a, z; \tau)]$ is monotone non-decreasing in $\eta$ (Lemma A.5). Based on the upper-tail cost value, we can obtain the risk-sensitive action using $\pi_\eta^\varphi$ (More details could be found in Appendix C).

For a given latent context variable $z$, the cost reduction achieved by the risk-sensitive policy compared to the risk-neutral policy is defined as:

$$
\Delta_c(\eta, z) := J_c(\pi_{exp}^\varphi | z) - J_c(\pi_\eta^\varphi | z).
\tag{26}
$$

*Table 1.* Performance comparison under training and deployment across tasks. D.R. denotes Domain Randomization, Rew. denotes reward, and O.R. denotes oscillation ratio, which is defined as the ratio of the AV's speed standard deviation to that of the preceding HDV over an episode, where a lower value indicates a greater capability of damping oscillations.

| Task | | Metric | Ours | Nominal | D.R. | SPiDR | Robust |
|---|---|---|---|---|---|---|---|
| PointGoal2 | Train | Rew. | 35.17±3.94 | 92.98±2.07 | 61.20±7.22 | 35.45±3.67 | 30.20±5.57 |
| | | Cost | 9.46±0.51 | 9.70±0.92 | 9.63±1.09 | 9.95±1.00 | 9.76±1.22 |
| | Deploy | Rew. | 49.59±8.52 | 91.43±7.25 | 83.17±10.79 | 43.44±4.14 | 12.90±9.32 |
| | | Cost | **8.86±0.85** | 29.62±3.73 | 35.92±5.45 | 11.20±1.65 | 24.07±2.55 |
| Autonomous Driving | Train | Rew. | 434.10±67.06 | 477.70±55.44 | 455.19±67.43 | 441.34±75.59 | 434.81±82.98 |
| | | Cost | 14.00±1.27 | 19.27±0.94 | 15.38±1.41 | 15.55±1.88 | 21.03±1.72 |
| | | O.R. | 0.43±0.13 | 0.45±0.12 | 0.61±0.133 | 0.45±0.09 | 0.50±0.10 |
| | | Jerk | 2.96±1.77 | 3.23±0.62 | 4.85±1.65 | 3.45±0.72 | 4.06±1.98 |
| | | Fuel | 0.316±0.029 | 0.311±0.005 | 0.341±0.131 | 0.316±0.006 | 0.324±0.074 |
| | Deploy | Rew. | 364.85±160.74 | 284.44±275.72 | 405.28±175.50 | 279.88±200.05 | 365.99±181.04 |
| | | Cost | **16.80±3.33** | 30.98±3.99 | 21.58±4.47 | 18.53±4.15 | 27.80±4.00 |
| | | O.R. | **0.47±0.13** | 0.50±0.34 | 0.60±0.15 | 0.47±0.18 | 0.53±0.15 |
| | | Jerk | **2.86±1.90** | 3.49±3.88 | 3.71±1.73 | 3.62±2.03 | 4.51±1.59 |
| | | Fuel | **0.314±0.026** | 0.354±0.053 | 0.321±0.019 | 0.322±0.028 | 0.327±0.019 |

Since the policy refinement in the deployment stage explicitly targets a more conservative upper-tail cost evaluation (Lemma A.5) and enforces more conservative action selection (Appendix C), the resulting policy $\pi_\eta^\varphi$ provides a non-negative cost reduction, i.e., $\Delta_c(\eta, z) \geq 0$. Then, we have the following theorem that ensures *Sim2Real* safety as shown below.

**Theorem 5.3** (Conditional *Sim2Real* safety under risk-sensitive deployment). *When the risk-sensitive policy $\pi_\eta^\varphi$ is deployed in a real world environment under Sim2Real gap, the cost constraint can be satisfied if $\Delta_c(\eta, z) \geq L_c\epsilon(N_{real})$.*

### 5.4. Dynamic Adaptation of Risk-sensitive Policy

In the deployment process, with more context collected, the estimation of the latent context variable $\hat{z}_{\xi^*}$ becomes more accurate. Accordingly, the *Sim2Real* gap decreases over time. As a result, the value of $\eta$ should be also be adjusted dynamically to maintain safety while improving performance (i.e., less conservative). This dynamic adaptation enables safe start at the early stage of the policy transfer, and efficient operation after the agent's behavior has converged.

We have the following theorem regarding existence of a feasible quantile $\eta$ for safe deployment.

**Theorem 5.4** (Existence of a safe deployment quantile). *There exists an integer $N_{\min} \geq 0$ such that for all $N_{real} >$*

$N_{\min}$, *one can choose a quantile $\eta^{(N_{real})} \in (0, 1)$ satisfying*

$$\Delta_c(\eta^{(N_{real})}, \hat{z}_{\xi^*}^{(N_{real})}) \geq L_c\,\epsilon(N_{real}). \quad (27)$$

According to Theorem 5.3 and 5.4, a desired $\eta$ that ensures safety is obtainable after collecting $N_{real} > N_{\min}$ contexts. To achieve dynamically adaptation of $\eta$ during deployment, an offline simulation calibration method is developed. Specifically, the relationships between $\epsilon$ and $N_{real}$, $L_c$ and $N_{real}$, as well as $\Delta_c$ and $\eta$, are calibrated offline using extensive simulations. Details of this calibration process are provided in Appendix D. Therefore, given $N_{real}$ contexts with corresponding $\eta^{(N_{real})}$, the system would be safe in expectation. As more real world contexts are collected, the estimated $\hat{z}_{\xi^*}$ is approaching $z^*$. Then the value of $\eta$ can be decreased gradually. Eventually, the risk-sensitive policy becomes a risk-neutral policy.

*Remark* 5.5. At the beginning of deployment, when no real context is available, the prior $p(z) = \mathcal{N}(\mathbf{0}_{d_z}, I_{d_z})$ is used to sample the latent context variable. The initial value of $\eta$ is obtained from the offline calibration results described above.

## 6. Experiment

The experiments are conducted on two tasks: 1) the Point-Goal2 task from the well-established benchmark OpenAI Safety Gym (Ray et al., 2019), and 2) a typical Autonomous

Driving task. More details about the experimental scenarios and the settings can be found in Appendix E. The number of training environments is 2048 with various random seeds, and the evaluation is conducted on 64 different new environments with 10 random seeds for each one, which are generated from different out-of-distribution (OOD) levels, as shown in Table 2.

**Baselines.** We compare our proposed method with several baselines in both tasks: (i) Nominal is a simple baseline that trains the agent only under the nominal condition; (ii) Domain Randomization trains the agent under perturbed parameters sampled from the training distribution; (iii) SPiDR uses a provable pessimistic safety upper bound during training so that the action is more conservative during deployment (As et al., 2025); and (iv) Robust RL adopts adversarial training under carefully designed observational perturbations, such that the worst-case trajectories could be optimized (Liu et al., 2023). All experiments are conducted using IQN (Dabney et al., 2018a) and Lagrangian (Stooke et al., 2020) methods to solve the CMDP problems, unless otherwise specified.

**Overall performance.** The overall training and deployment results are summarized in Table 1. During training, all methods successfully learn to reach the goal without violating the cost constraints (i.e., 10 in POINTGOAL2 task and 20 in Autonomous Driving task). In deployment, however, a clear performance divergence emerges. Nominal, Domain Randomized, and Robust RL baselines achieve high rewards but suffer from severe cost violations, indicating limited safety in the deployment environment with *Sim2Real* gap. In contrast, both SPiDR and our method exhibit conservative behaviors and have substantially lower costs, at the expense of lower rewards. This reward-cost trade-off is expected in both tasks, where more cautious navigation leads to better hazards avoidance in unseen environments in the first task, and more conservative AV speed-control reduce the risk of rear-end collisions in the second task. Notably, compared to SPiDR, our method achieves a more favorable trade-off by attaining lower deployment costs while simultaneously achieving higher rewards. Besides, in the Autonomous Driving task, our method also shows strong oscillation-reduction (O.R.) effects, reducing O.R., jerk, and fuel consumption from $0.922\pm0.017$, $12.329\pm0.195$, and $0.440\pm0.008$ under the default FVD controller (Jiang et al., 2001) to $0.47\pm0.13$, $2.86\pm1.90$, and $0.314\pm0.026$ under deployment.

**Dynamic risk-sensitive adaptation.** Figure 3 (a) illustrates the change of cost value over collected contexts in the deployment stage using different methods in the POINTGOAL2 task. Our method (blue curve) consistently maintains the cost below the threshold throughout the entire deployment process, demonstrating a clear safety advantage

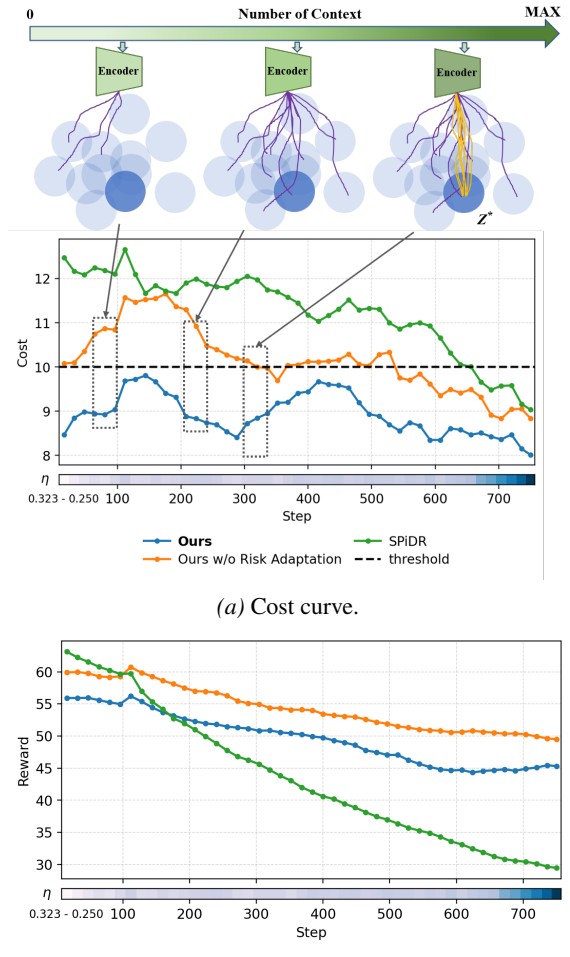

*(a)* Cost curve.

*(b)* Reward curve.

*Figure 3.* Deployment performance under dynamic risk-sensitive adaptation in POINTGOAL2 task.

over other baselines. The safety guaranty is achieved by the proposed dynamic risk-sensitive adaptation module, which actively regulates the policy during deployment. When this adaptation module is removed and agent starts with the risk-neural policy, the cost (orange curve) initially exceeds the threshold due to limited real world context, and then gradually decreases as additional interactions improve the latent context variable estimation (i.e., $\hat{z}_{\xi^*}^{(N_{\text{real}})}$).

This comparison shows that the dynamic risk-sensitive adaptation module accommodates the initial uncertainty by enforcing conservative behavior in the early deployment stage, thereby ensuring safety until sufficient context has been collected. The evolution of the risk parameter $\eta$ further reflects this adaptation process, transitioning from a risk-sensitive to a more risk-neutral policy as the context number increases. Note that $\eta$ does not decrease to zero in the experiments, as the cost discrepancy $\Delta_c$ approaches zero around $\eta \approx 0.25$, which is also consistent with the offline calibration results.

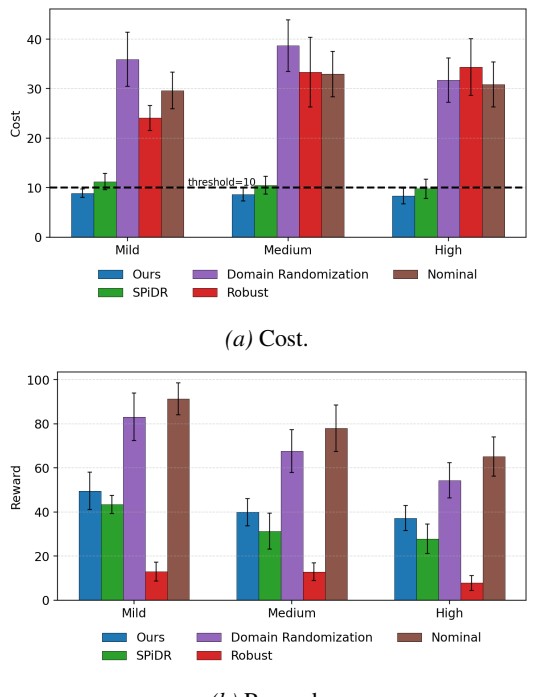

*(a)* Cost.

*(b)* Reward.

*Figure 4.* Performance under mild, medium, and high OOD level scenarios in POINTGOAL2 task.

Meanwhile, the reward curves in Figure 3 (b) show that our method maintains high reward values despite operating under a risk-sensitive policy. In contrast, under SPiDR, the agent may adopt an overly conservative behavior by completely stopping the agent to avoid cost violations, which reduces both the accumulated reward and the incurred cost. A demonstration video of agent behaviors under different methods is provided in Appendix F.

**OOD sensitivity analysis.** To evaluate the robustness of the proposed model under different distributional shifts, we conduct a sensitivity analysis with regard to different OOD levels, with detailed settings in Appendix E. As shown in Figure 4, increasing OOD level generally leads to higher costs and lower rewards. Nevertheless, our method consistently maintains deployment costs below the threshold across all OOD scenarios.

The ablation study, online computational overhead and complexity analysis could be found in in Appendix F.

## 7. Conclusions and Discussions

This work presents a transferable reinforcement learning framework that enables policies trained in simulation to be deployed in real-world environments with safety assurance. By integrating probabilistic latent context embedding with distributional reinforcement learning, the proposed framework explicitly addresses safety violations caused by environment mismatch, known as the *Sim2Real* gap. Through dynamic risk-sensitive policy adaptation with limited real-world interactions and a principled safety upper bound, the framework provably ensures safe deployment while gradually relaxing risk-sensitive policies as the latent context estimation becomes more accurate. Extensive experiments on the POINTGOAL2 task and Autonomous Driving task validate the effectiveness of the proposed approach. Across training, deployment, and multiple OOD scenarios, the method consistently maintains deployment costs below the threshold while achieving competitive rewards. In contrast to baseline methods, which either violate safety constraints or adopt overly conservative behaviors that degrade task performance, our method achieves a more favorable reward-cost trade-off.

Offline RL is another useful backbone to solve these safety-critical tasks. In principle, the proposed framework can also be implemented in the offline RL context using trajectories from real world environments. However, because collecting real-world trajectories is costly, the amount of available data is often limited and biased. As a result, both the encoder of the context variable $z$ and the estimates of the value function distribution $Z_r$, $Z_c$ may be inaccurate. In particular, limited coverage of system dynamics can lead to poor estimation of the context variable, while the lack of edge-case samples can result in inaccurate estimation of the tails of the value function distributions.

For future work, evaluating the proposed framework in a broader range of tasks could further assess its generality and robustness. Besides, in this work, we mainly consider adaptation to an unknown but fixed environment during deployment. Handling continuously changing environments is an interesting topic. One possible direction is to design a detection mechanism for changes in the environment, such as a sudden shift from dry pavement to icy pavement. By integrating the detection module with our framework, the new method could better handle dynamically changing environments.

## Acknowledgments

This research is supported in part by the U.S. National Science Foundation (NSF) through Grant CMMI #2339753. We sincerely thank the reviewers for their constructive comments, which helped improve the quality of this work.

## Impact Statement

This paper presents work whose goal is to advance the field of Machine Learning. There are many potential societal consequences of our work, none which we feel must be specifically highlighted here.

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

# Appendix

# A. Proofs

### A.1. Preliminaries

**Definition A.1** (Latent context estimation error). In the real environment with $\xi^*$, the latent context estimation from $N_{real}$ context samples is $\hat{z}_{\xi^*} = \mathbb{E}_{q_\phi(z|D_{\xi^*}^{N_{real}})}[z]$. With the real $z^*$, the latent context estimation error is defined as:

$$\epsilon(N_{real}) := \mathbb{E}[||\hat{z}_{\xi^*} - z^*||_2], \tag{28}$$

with $\epsilon(N_{real}) \to 0$ as $N_{real} \to \infty$. The encoder's mean output approaches a stable latent representation $z^*$ for the real environment as more real data is collected.

**Assumption A.2** (Lipschitz continuity). For any policy $\pi$ in any environment with $\xi$, the expected cost function $J_c(\pi|\xi)$ is $L_c$-Lipschitz continuous. Therefore, for any two latent context $z_1$ and $z_2$,

$$|J_c(\pi|z_1) - J_c(\pi|z_2)| \le L_c||z_1 - z_2||_2, \tag{29}$$

which captures that the CMDP cost changes smoothly as the environment latent changes.

**Assumption A.3** (Training safety guarantee). The Lagrangian training in simulation yields a risk-neutral policy $\pi_{exp}^\varphi$ that satisfies the safety bound $J_c(\pi_{exp}^\varphi|\hat{z}) \le d$.

**Assumption A.4** (Probabilistic calibration validity). All simulation-calibrated quantities $\epsilon_{\text{sim}}(N)$, $L_{c,\text{sim}}$, and $\Delta_{\text{sim}}(\eta)$ upper/lower bound their real counterparts with probability at least $q_{\text{safe}} := \min\{q_\epsilon, q_L, 1 - q_\Delta\}$.

More precisely, for any $N \ge 0$,

$$\mathbb{P}\Big(\|\hat{z}_{\xi^*}^{(N)} - z^*\|_2 \le \epsilon_{\text{sim}}(N)\Big) \ge q_\epsilon, \tag{30}$$

$$\mathbb{P}\Big(|J_c(\pi|z^*) - J_c(\pi|\hat{z}_{\xi^*}^{(N)})| \le L_{c,\text{sim}} \epsilon_{\text{sim}}(N)\Big) \ge q_L, \tag{31}$$

and for any $\eta \in (0,1)$,

$$\mathbb{P}(\Delta(\eta, z^*) \ge \Delta_{\text{sim}}(\eta)) \ge 1 - q_\Delta. \tag{32}$$

**Lemma A.5** (Monotonicity of upper-tail cost value). *Fix $(s, a, z)$ and consider the IQN-based cost critic $Z^{\theta_c}(s, a, z; \tau)$. Define the upper-tail cost value*

$$Q_c^\eta(s, a, z) := \mathbb{E}_{\tau \sim U([\eta,1])}\big[Z^{\theta_c}(s, a, z; \tau)\big], \quad \eta \in (0, 1).$$

*For any $0 < \eta_1 \le \eta_2 < 1$, it holds that*

$$Q_c^{\eta_1}(s, a, z) \le Q_c^{\eta_2}(s, a, z).$$

### A.2. Proof of Lemma 4.1 (Latent-conditioned contraction)

*Proof.* Fix any $(s, a)$ and the same latent $z$. Let $s' \sim p(\cdot|s, a, z)$. For $i \in \{1, 2\}$, define the Bellman target random return

$$Y_i := r(s, a, z) + \gamma X_i, \tag{33}$$

where $X_i \sim Z_i(s', \pi(s', z), z)$. By definition, the distribution of $Y_i$ equals $(\mathcal{T}_r^{\pi, z} Z_i)(s, a)$.

First sample $s' \sim p(\cdot|s, a, z)$. Conditioned on this realized $s'$, take an *optimal* coupling $(X_1, X_2)$ between the two distributions $Z_1(s', \pi(s', z), z)$ and $Z_2(s', \pi(s', z), z)$. This defines a valid coupling of $(Y_1, Y_2)$ with correct marginals.

By the definition of Wasserstein distance as the infimum over all couplings, for our particular coupling we have

$$W_p^p((\mathcal{T}_r^{\pi, z} Z_1)(s, a), (\mathcal{T}_r^{\pi, z} Z_2)(s, a)) \le \mathbb{E}[|Y_1 - Y_2|^p]. \tag{34}$$

Since $Y_1 - Y_2 = \gamma(X_1 - X_2)$ and $r(s, a, z)$ cancels out,

$$\mathbb{E}[|Y_1 - Y_2|^p] = \gamma^p \mathbb{E}[|X_1 - X_2|^p]. \tag{35}$$

By optimality of the conditional coupling of $(X_1, X_2)$ given $s'$,

$$\mathbb{E}[|X_1 - X_2|^p \mid s'] = W_p^p(Z_1(s', \pi(s', z), z), \; Z_2(s', \pi(s', z), z)). \tag{36}$$

Taking expectation over $s'$ and using the supremum definition of $\bar{d}_p$,

$$\mathbb{E}[|X_1 - X_2|^p] \le \bar{d}_p^p(Z_1, Z_2). \tag{37}$$

Plugging into (34) and taking the $p$-th root yields

$$W_p((\mathcal{T}_r^{\pi,z} Z_1)(s, a), \; (\mathcal{T}_r^{\pi,z} Z_2)(s, a)) \le \gamma \, \bar{d}_p(Z_1, Z_2). \tag{38}$$

Finally take $\sup_{s,a}$ on both sides to obtain (15). The cost case is identical by replacing $r$ with $c$. $\qquad\square$

## A.3. Proof of Theorem 4.2 (Adding latent context $z$ preserves Bellman contraction and fixed-point well-posedness)

*Proof.* The $\gamma$-contraction property follows directly from Lemma 4.1. By the Banach fixed-point theorem (Kreyszig, 1991), a contraction mapping on a complete metric space admits a unique fixed point, and repeated application of the operator converges to it at rate $\gamma^k$ under the same metric $\bar{d}_p$. The cost case is identical. $\qquad\square$

## A.4. Proof of Lemma 4.3 (Descent lemma (smooth upper bound))

*Proof.* Define $g(t) := \mathcal{J}(x + t\Delta)$ for $t \in [0, 1]$. Then $g'(t) = \langle \nabla \mathcal{J}(x + t\Delta), \Delta \rangle$ and

$$\mathcal{J}(x + \Delta) - \mathcal{J}(x) = \int_0^1 g'(t) \, dt = \int_0^1 \langle \nabla \mathcal{J}(x + t\Delta), \Delta \rangle dt. \tag{39}$$

Add and subtract $\nabla \mathcal{J}(x)$ inside the inner product and apply Cauchy–Schwarz:

$$\mathcal{J}(x + \Delta) = \mathcal{J}(x) + \langle \nabla \mathcal{J}(x), \Delta \rangle + \int_0^1 \langle \nabla \mathcal{J}(x + t\Delta) - \nabla \mathcal{J}(x), \Delta \rangle dt \tag{40}$$

$$\le \mathcal{J}(x) + \langle \nabla \mathcal{J}(x), \Delta \rangle + \int_0^1 \|\nabla \mathcal{J}(x + t\Delta) - \nabla \mathcal{J}(x)\|_2 \, \|\Delta\|_2 \, dt. \tag{41}$$

Using $L$-Lipschitz gradients, $\|\nabla \mathcal{J}(x + t\Delta) - \nabla \mathcal{J}(x)\|_2 \le Lt\|\Delta\|_2$, thus

$$\mathcal{J}(x + \Delta) \le \mathcal{J}(x) + \langle \nabla \mathcal{J}(x), \Delta \rangle + \int_0^1 Lt\|\Delta\|_2^2 \, dt = \mathcal{J}(x) + \langle \nabla \mathcal{J}(x), \Delta \rangle + \frac{L}{2}\|\Delta\|_2^2. \tag{42}$$

$$\square$$

## A.5. Proof of Theorem 4.4 (Alternating frozen training is stable under constant step sizes)

*Proof.* Define the stacked parameter vector

$$x_k := (\theta_{r,k}, \theta_{c,k}, \phi_k), \tag{43}$$

and let $\mathcal{F}_k$ denote the $\sigma$-algebra generated by all randomness up to iteration $k$ (e.g., past mini-batches / replay-buffer samples, environment transitions, and the current parameters). Assume the stochastic gradients are conditionally unbiased with bounded conditional second moments:

$$\mathbb{E}\left[\widehat{\nabla}_{\theta_r} \mathcal{L}_r^{\text{critic}}(\theta_{r,k}; \phi_k) \mid \mathcal{F}_k\right] = \nabla_{\theta_r} \mathcal{L}_r^{\text{critic}}(\theta_{r,k}; \phi_k), \qquad \mathbb{E}\left[\left\|\widehat{\nabla}_{\theta_r} \mathcal{L}_r^{\text{critic}}(\theta_{r,k}; \phi_k)\right\|_2^2 \mid \mathcal{F}_k\right] \le G_r^2, \tag{44}$$

and analogously for $\widehat{\nabla}_{\theta_c} \mathcal{L}_c^{\text{critic}}$ and $\widehat{\nabla}_\phi \mathcal{L}^{\text{encoder}}$ with bounds $G_c^2$ and $G_\phi^2$.

Recall the potential function

$$\mathcal{J}(\theta_r, \theta_c, \phi) := \beta_r \mathcal{L}_r^{\text{critic}}(\theta_r; \phi) + \beta_c \mathcal{L}_c^{\text{critic}}(\theta_c; \phi) + \beta_{KL} \mathcal{L}_{KL}(\phi). \tag{45}$$

**(I) Critic step decrease (encoder frozen).** Define the intermediate iterate $\tilde{x}_{k+1} := (\theta_{r,k+1}, \theta_{c,k+1}, \phi_k)$ after the critic update. Apply Lemma 4.3 to the mapping $(\theta_r, \theta_c) \mapsto \mathcal{J}(\theta_r, \theta_c, \phi_k)$ with update

$$\Delta_k^{\theta_r} := \theta_{r,k+1} - \theta_{r,k} = -\alpha \widehat{\nabla}_{\theta_r} \mathcal{L}_r^{\text{critic}}(\theta_{r,k}; \phi_k), \qquad \Delta_k^{\theta_c} := \theta_{c,k+1} - \theta_{c,k} = -\alpha \widehat{\nabla}_{\theta_c} \mathcal{L}_c^{\text{critic}}(\theta_{c,k}; \phi_k). \tag{46}$$

Then there exists $L_\theta > 0$ such that

$$\mathcal{J}(\tilde{x}_{k+1}) \le \mathcal{J}(x_k) + \left\langle \nabla_{\theta_r} \mathcal{J}(x_k), \Delta_k^{\theta_r} \right\rangle + \left\langle \nabla_{\theta_c} \mathcal{J}(x_k), \Delta_k^{\theta_c} \right\rangle + \frac{L_\theta}{2} \left( \|\Delta_k^{\theta_r}\|_2^2 + \|\Delta_k^{\theta_c}\|_2^2 \right). \tag{47}$$

Taking conditional expectation given $\mathcal{F}_k$ and using (44) yields

$$\mathbb{E}[\mathcal{J}(\tilde{x}_{k+1}) \mid \mathcal{F}_k] \le \mathcal{J}(x_k) - \alpha \left( \|\nabla_{\theta_r} \mathcal{J}(x_k)\|_2^2 + \|\nabla_{\theta_c} \mathcal{J}(x_k)\|_2^2 \right) + \frac{L_\theta \alpha^2}{2} \left( G_r^2 + G_c^2 \right). \tag{48}$$

**(II) Encoder step decrease (critics frozen).** Now perform the encoder update to obtain $x_{k+1} = (\theta_{r,k+1}, \theta_{c,k+1}, \phi_{k+1})$. Apply Lemma 4.3 to the mapping $\phi \mapsto \mathcal{J}(\theta_{r,k+1}, \theta_{c,k+1}, \phi)$ with update

$$\Delta_k^\phi := \phi_{k+1} - \phi_k = -\beta \widehat{\nabla}_\phi \mathcal{L}^{\text{encoder}}(\phi_k; \theta_{r,k+1}, \theta_{c,k+1}). \tag{49}$$

Then there exists $L_\phi > 0$ such that

$$\mathcal{J}(x_{k+1}) \le \mathcal{J}(\tilde{x}_{k+1}) + \left\langle \nabla_\phi \mathcal{J}(\tilde{x}_{k+1}), \Delta_k^\phi \right\rangle + \frac{L_\phi}{2} \|\Delta_k^\phi\|_2^2. \tag{50}$$

Taking conditional expectation given $\mathcal{F}_k$ and using (44) yields

$$\mathbb{E}[\mathcal{J}(x_{k+1}) \mid \mathcal{F}_k] \le \mathbb{E}[\mathcal{J}(\tilde{x}_{k+1}) \mid \mathcal{F}_k] - \beta \|\nabla_\phi \mathcal{J}(\tilde{x}_{k+1})\|_2^2 + \frac{L_\phi \beta^2}{2} G_\phi^2. \tag{51}$$

**(III) Combine.** Combining (48) and (51) gives

$$\mathbb{E}[\mathcal{J}(x_{k+1}) \mid \mathcal{F}_k] \le \mathcal{J}(x_k) - \alpha \left( \|\nabla_{\theta_r} \mathcal{J}(x_k)\|_2^2 + \|\nabla_{\theta_c} \mathcal{J}(x_k)\|_2^2 \right) - \beta \|\nabla_\phi \mathcal{J}(\tilde{x}_{k+1})\|_2^2$$
$$+ \frac{L_\theta \alpha^2}{2} (G_r^2 + G_c^2) + \frac{L_\phi \beta^2}{2} G_\phi^2. \tag{52}$$

Define the constant

$$C := \max \left\{ \tfrac{L_\theta}{2} (G_r^2 + G_c^2), \ \tfrac{L_\phi}{2} G_\phi^2 \right\} < \infty. \tag{53}$$

Then (52) implies the compact bound

$$\mathbb{E}[\mathcal{J}(x_{k+1}) \mid \mathcal{F}_k] \le \mathcal{J}(x_k) - \alpha \left( \|\nabla_{\theta_r} \mathcal{J}(x_k)\|_2^2 + \|\nabla_{\theta_c} \mathcal{J}(x_k)\|_2^2 \right) - \beta \|\nabla_\phi \mathcal{J}(\tilde{x}_{k+1})\|_2^2 + C(\alpha^2 + \beta^2). \tag{54}$$

**(IV) Telescope and take** $\lim \sup$**.** Take total expectation and use the tower property to obtain

$$\mathbb{E}[\mathcal{J}(x_{k+1})] \le \mathbb{E}[\mathcal{J}(x_k)] - \alpha \, \mathbb{E}\left[\|\nabla_{\theta_r} \mathcal{J}(x_k)\|_2^2 + \|\nabla_{\theta_c} \mathcal{J}(x_k)\|_2^2\right] - \beta \, \mathbb{E}\left[\|\nabla_\phi \mathcal{J}(\tilde{x}_{k+1})\|_2^2\right] + C(\alpha^2 + \beta^2). \tag{55}$$

Summing (55) from $k = 0$ to $K - 1$ yields

$$\alpha \sum_{k=0}^{K-1} \mathbb{E}\left[\|\nabla_{\theta_r} \mathcal{J}(x_k)\|_2^2 + \|\nabla_{\theta_c} \mathcal{J}(x_k)\|_2^2\right] + \beta \sum_{k=0}^{K-1} \mathbb{E}\left[\|\nabla_\phi \mathcal{J}(\tilde{x}_{k+1})\|_2^2\right] \le \mathbb{E}[\mathcal{J}(x_0)] - \mathbb{E}[\mathcal{J}(x_K)] + CK(\alpha^2 + \beta^2). \tag{56}$$

Using the lower bound $\mathcal{J}(x) \ge \mathcal{J}_{\text{inf}}$ and dividing by $K$ gives

$$\frac{\alpha}{K} \sum_{k=0}^{K-1} \mathbb{E}\left[\|\nabla_{\theta_r} \mathcal{J}(x_k)\|_2^2 + \|\nabla_{\theta_c} \mathcal{J}(x_k)\|_2^2\right] + \frac{\beta}{K} \sum_{k=0}^{K-1} \mathbb{E}\left[\|\nabla_\phi \mathcal{J}(\tilde{x}_{k+1})\|_2^2\right] \le \frac{\mathbb{E}[\mathcal{J}(x_0)] - \mathcal{J}_{\text{inf}}}{K} + C(\alpha^2 + \beta^2). \tag{57}$$

Let $K \to \infty$ and use $\alpha, \beta \le \max\{\alpha, \beta\}$ to obtain

$$\limsup_{K \to \infty} \frac{1}{K} \sum_{k=0}^{K-1} \mathbb{E}\big[\|\nabla_{\theta_r} \mathcal{J}(x_k)\|_2^2 + \|\nabla_{\theta_c} \mathcal{J}(x_k)\|_2^2 + \|\nabla_\phi \mathcal{J}(\tilde{x}_{k+1})\|_2^2\big] \le O(\max\{\alpha, \beta\}). \tag{58}$$

Finally, since

$$\|\nabla \mathcal{J}(\theta_{r,k}, \theta_{c,k}, \phi_k)\|_2^2 = \|\nabla_{\theta_r} \mathcal{J}(x_k)\|_2^2 + \|\nabla_{\theta_c} \mathcal{J}(x_k)\|_2^2 + \|\nabla_\phi \mathcal{J}(x_k)\|_2^2,$$

and $\|\nabla_\phi \mathcal{J}(x_k)\|_2^2$ differs from $\|\nabla_\phi \mathcal{J}(\tilde{x}_{k+1})\|_2^2$ only by a higher-order term controlled by smoothness and the $O(\alpha)$ critic step, we conclude (23). $\qquad\square$

### A.6. Proof of Proposition 4.5 (Irreducible regret under task aliasing without $z$)

*Proof.* Let $p := \pi(\boldsymbol{a}_1|\bar{\boldsymbol{s}})$, so $\pi(\boldsymbol{a}_2|\bar{\boldsymbol{s}}) = 1 - p$. In $\xi_1$, the expected regret at $\bar{\boldsymbol{s}}$ is at least

$$\underbrace{(1 - p)}_{\text{wrong prob.}} \underbrace{\left(Q_r^{\xi_1}(\bar{\boldsymbol{s}}, \boldsymbol{a}_1) - Q_r^{\xi_1}(\bar{\boldsymbol{s}}, \boldsymbol{a}_2)\right)}_{\text{wrong loss}} \ge (1 - p)\delta. \tag{59}$$

In $\xi_2$, the expected regret at $\bar{\boldsymbol{s}}$ is at least

$$\underbrace{p}_{\text{wrong prob.}} \underbrace{\left(Q_r^{\xi_2}(\bar{\boldsymbol{s}}, \boldsymbol{a}_2) - Q_r^{\xi_2}(\bar{\boldsymbol{s}}, \boldsymbol{a}_1)\right)}_{\text{wrong loss}} \ge p\delta. \tag{60}$$

Therefore, $\max\{p\delta, (1 - p)\delta\} \ge \delta/2$, meaning that at least one of $\xi_1$ or $\xi_2$ incurs regret $\ge \delta/2$ at $\bar{\boldsymbol{s}}$, independent of data size. $\qquad\square$

### A.7. Proof of Theorem 5.2 (*Sim2Real* safety upper-bound under risk-neutral deployment)

*Proof.* By Lipschitz continuity (Assumption A.2), for any policy $\pi$,

$$|J_c(\pi|\boldsymbol{z}_1) - J_c(\pi|\boldsymbol{z}_2)| \le L_c\|\boldsymbol{z}_1 - \boldsymbol{z}_2\|_2. \tag{61}$$

With risk-neutral policy $\pi_{exp}^\varphi$,

$$|J_c(\pi_{exp}^\varphi|\boldsymbol{z}^*) - J_c(\pi_{exp}^\varphi|\hat{\boldsymbol{z}})| \le L_c\|\boldsymbol{z}^* - \hat{\boldsymbol{z}}\|_2. \tag{62}$$

Then,

$$J_c(\pi_{exp}^\varphi|\hat{\boldsymbol{z}}) \le J_c(\pi_{exp}^\varphi|\boldsymbol{z}^*) + L_c\|\boldsymbol{z}^* - \hat{\boldsymbol{z}}\|_2. \tag{63}$$

By training safety (Assumption A.3) and real environment characteristic (Assumption 5.1), for all latent including $\boldsymbol{z}^*$, we have

$$J_c(\pi_{exp}^\varphi|\boldsymbol{z}^*) \le d. \tag{64}$$

Thus,

$$J_c(\pi_{exp}^\varphi|\hat{\boldsymbol{z}}) \le d + L_c\|\boldsymbol{z}^* - \hat{\boldsymbol{z}}\|_2. \tag{65}$$

Taking expectation over the randomness in $\hat{\boldsymbol{z}}$,

$$J_c(\pi_{exp}^\varphi|\hat{\boldsymbol{z}}) \le d + L_c\epsilon(N_{real}). \tag{66}$$

$$\square$$

### A.8. Proof of Theorem 5.3 (Conditional *Sim2Real* safety under risk-sensitive deployment)

*Proof.* Like proof A.7, the deployed policy satisfies:

$$|J_c(\pi_\eta^\varphi|\boldsymbol{z}^*) - J_c(\pi_\eta^\varphi|\hat{\boldsymbol{z}})| \le L_c\|\boldsymbol{z}^* - \hat{\boldsymbol{z}}\|_2. \tag{67}$$

By definition of Eq. (26),

$$J_c(\pi_\eta^\varphi|\hat{\boldsymbol{z}}) \le J_c(\pi_{exp}^\varphi|\boldsymbol{z}^*) - \Delta(\eta, \hat{\boldsymbol{z}}) + L_c\|\boldsymbol{z}^* - \hat{\boldsymbol{z}}\|_2. \tag{68}$$

Using upper-bound in Assumption A.3,

$$J_c(\pi_\eta^\varphi|\hat{z}) \leq d - \Delta(\eta, \hat{z}) + L_c||z^* - \hat{z}||_2. \tag{69}$$

Therefore, if $\Delta(\eta, z) \geq L_c\epsilon(N_{real}) = L_c\mathbb{E}||z^* - \hat{z}||_2$, then the RHS is at most $d$, which yields the claimed safety condition.

$\square$

### A.9. Proof of Lemma A.5 (Monotonicity of upper-tail cost value)

*Proof.* Fix $(s, a, z)$ and denote $f(\tau) := Z^{\theta_c}(s, a, z; \tau)$. The quantile function $f(\tau)$ is non-decreasing in $\tau$. For $\eta \in (0, 1)$, let $\tau = \eta + (1 - \eta)u$ with $u \sim U([0, 1])$. Then

$$Q_c^\eta(s, a, z) = \mathbb{E}_{u\sim U([0,1])}[f(\eta + (1 - \eta)u)].$$

For any $0 < \eta_1 \leq \eta_2 < 1$ and any $u \in [0, 1]$,

$$\eta_1 + (1 - \eta_1)u \leq \eta_2 + (1 - \eta_2)u,$$

since their difference equals $(\eta_2 - \eta_1)(1 - u) \geq 0$. By the monotonicity of $f$, we have $f(\eta_1 + (1-\eta_1)u) \leq f(\eta_2 + (1-\eta_2)u)$ for all $u$. Taking expectation over $u$ yields $Q_c^{\eta_1}(s, a, z) \leq Q_c^{\eta_2}(s, a, z)$. $\square$

### A.10. Proof of Theorem 5.4 (Existence of a safe deployment quantile)

*Proof.* Fix any quantile $\eta \in (0, 1)$, $\Delta_c(\eta) \geq 0$. Since $\epsilon(N) \to 0$ as $N \to \infty$, by the $\varepsilon-N$ definition of limits in $\mathbb{R}$, for the positive constant

$$\varepsilon := \frac{\Delta_c(\eta)}{L_c}, \tag{70}$$

there exists an integer $N_{\min} \geq 0$ such that for all $N_{real} > N_{\min}$,

$$\left|\epsilon(N_{real}) - 0\right| < \varepsilon = \frac{\Delta_c(\eta)}{L_c}. \tag{71}$$

Multiplying both sides by $L_c > 0$ yields

$$L_c\,\epsilon(N_{real}) < \Delta_c(\eta), \quad \forall N_{real} > N_{\min}. \tag{72}$$

Now define the deployment quantile for all $N_{real} > N_{\min}$ as

$$\eta^{(N_{real})} := \eta. \tag{73}$$

Therefore,

$$\Delta_c(\eta^{(N_{real})}) = \Delta_c(\eta) > L_c\,\epsilon(N_{real}), \tag{74}$$

which proves that for every $N_{real} > N_{\min}$ there exists a feasible deployment quantile $\eta^{(N_{real})} \in (0, 1)$ satisfying the desired inequality. $\square$

## B. Lagrangian Training Process

Let $\hat{J}_c$ be the estimated discounted cost under the current policy. The constraint violation is defined as:

$$e = \hat{J}_c(\pi) - d. \tag{75}$$

The PID controller generates an update signal:

$$\Delta\lambda_L = K_p e + K_i \sum_j e_j + K_d(e - e_{\text{prev}}), \tag{76}$$

where $K_p$, $K_i$, and $K_d$ are proportional, integral and derivative gains. Finally, the multiplier is updated with non-negative projection:

$$\lambda_L \leftarrow \max\left(0, \lambda_L + \Delta\lambda_L\right). \tag{77}$$

This adaptive update increases $\lambda_L$ when the constraint is violated and decreases it when the policy becomes overly conservative, thus stabilizing the Lagrangian optimization.

## C. Risk-Aware Conservative Action Generation at Deployment

To construct the risk-sensitive deployment policy $\pi_\eta^\varphi$, a lightweight refinement of the nominal actor output is conducted using the learned IQN critics, without updating any network parameters. The method is summarized in Algorithm 1.

Specifically, the upper-tail cost value is

$$Q_c^\eta(s, a, z) := \mathbb{E}_{\tau \sim U([\eta, 1])}\big[Z^{\theta_c}(s, a, z; \tau)\big], \quad \eta \in (0, 1). \tag{78}$$

Let the nominal action produced by the trained actor be

$$a_0 = \pi_{exp}^\varphi(s, z). \tag{79}$$

At deployment, $a_0$ is refined to obtain a conservative action that satisfies a risk-aware cost constraint while remaining close to the nominal behavior. Specifically, we solve a local constrained problem:

$$\begin{aligned} \max_{a \in \mathcal{A}} \quad & Q_r(s, a, z) - \beta\|a - a_0\|_2^2 \\ \text{s.t.} \quad & Q_c^\eta(s, a, z) \leq d, \end{aligned} \tag{80}$$

where $Q_r(s, a, z) := \mathbb{E}_{\tau \sim U([0,1])}[Z_r^{\theta_r}(s, a, z; \tau)]$ is the risk-neutral reward value, and $\beta_n > 0$ penalizes excessive deviation from the nominal action.

Problem (80) is approximately solved by a projected gradient refinement initialized at $a_0$. At iteration $k$, we first take a reward-ascent step with proximal regularization,

$$\tilde{a}_{k+1} = \Pi_\mathcal{A}\Big(a_k + \alpha_r\big(\nabla_a Q_r(s, a_k, z) - 2\beta_n(a_k - a_0)\big)\Big), \tag{81}$$

and then apply a risk-aware projection step to reduce the upper-tail cost value:

$$a_{k+1} = \Pi_\mathcal{A}\Big(\tilde{a}_{k+1} - \alpha_c \nabla_a Q_c^\eta(s, \tilde{a}_{k+1}, z)\Big), \tag{82}$$

where $\Pi_\mathcal{A}$ denotes Euclidean projection onto $\mathcal{A}$, and $\alpha_r, \alpha_c > 0$ are step sizes. The refinement terminates once $Q_c^\eta(s, a_k, z) \leq d$ or after a fixed number of iterations $K_{\text{ref}}$. If the nominal action already satisfies the constraint, i.e., $Q_c^\eta(s, a_0, z) \leq d$, we directly execute $a_0$.

Let $a_{\text{ref}}$ denote the refined action after the projected updates (either after $K_{\text{ref}}$ steps or early stop). We apply a refinement weight

$$w(\eta) = \text{clip}(\eta, 0, 1),$$

and execute the interpolated action

$$a_\eta = a_0 + w(\eta)\,(a_{\text{ref}} - a_0).$$

This ensures $\eta = 0$ yields the nominal action $a_0$ (risk-neutral), while $\eta = 1$ yields the fully refined action.

The resulting deployment action defines the risk-sensitive policy $\pi_\eta^\varphi$:

$$\pi_\eta^\varphi(s, z) := a_\eta. \tag{83}$$

*Remark* C.1. The refinement is performed only at deployment and does not modify the actor or critic parameters. The proximal term $\beta\|a - a_0\|_2^2$ helps prevent the action from drifting into out-of-distribution regions where critic gradients may be less reliable. In implementation, both $Q_r$ and $Q_c^\eta$ are estimated by averaging over a small set of quantile samples, and the gradients are computed by automatic differentiation.

## D. Offline Simulation-based Calibration Method

An offline simulation-based calibration method is introduced to estimate the latent sensitivity and the risk-sensitive return reduction.

---

**Algorithm 1** Risk-Aware Projected Action Refinement at Deployment

---

**Require:** State $s$, latent $z$, nominal actor $\pi_{exp}^{\varphi}$
**Require:** Reward critic $Z^{\theta_r}$, cost critic $Z^{\theta_c}$, risk level $\eta$
**Require:** Step sizes $\alpha_r, \alpha_c$, proximal weight $\beta_n$
**Ensure:** Conservative action $a_\eta$
 1: Initialize $a_0 \leftarrow \pi_{exp}^{\varphi}(s, z)$
 2: $a \leftarrow a_0$
 3: **for** $k = 1$ to $K_{\text{ref}}$ **do**
 4:     Evaluate reward value $Q_r(s, a, z)$ using uniform quantiles
 5:     Evaluate upper-tail cost $Q_c^\eta(s, a, z)$ using $\tau \sim U([\eta, 1])$
 6:     **if** $Q_c^\eta(s, a, z) \leq d$ **then**
 7:         **break**
 8:     **end if**
 9:     Reward-oriented update:
10:     $\tilde{a} \leftarrow \Pi_{\mathcal{A}}\big(a + \alpha_r\big(\nabla_a Q_r(s, a, z) - 2\beta_n(a - a_0)\big)\big)$
11:     Cost-aware projection:
12:     $a \leftarrow \Pi_{\mathcal{A}}(\tilde{a} - \alpha_c \nabla_a Q_c^\eta(s, \tilde{a}, z))$
13: **end for**
14: $a_\eta \leftarrow a_0 + w(\eta)\,(a - a_0)$
15: **return** $a_\eta$

---

**Latent sensitivity for $\epsilon$.** The true latent context $\boldsymbol{z}^*$ is never observable directly in simulations. However, $\boldsymbol{z}^*$ can be approximated using a large number of offline collected context. Specifically, for each simulated environment $\xi \sim \mu_{env}$, a set $D_\xi^{(N_{ref})}$ containing sufficient large context $N_{ref}$ is adopted, and then the reference latent $\hat{\boldsymbol{z}}_\xi^{ref}$ is estimated based on the contexts, which becomes a substitute for $\boldsymbol{z}^*$.

$$\hat{\boldsymbol{z}}_\xi^{ref} = \mathbb{E}_{q_\phi(\boldsymbol{z}|D_\xi^{(N_{ref})})}[\boldsymbol{z}] \approx \boldsymbol{z}^*. \tag{84}$$

To find the relationship between the number of context $N$ and encoder error $\epsilon$, a Monte Carlo method is used. Specifically, $N$ contexts are sampled to form $D_\xi^{(N)}$ and $\hat{\boldsymbol{z}}_\xi^{(N)}$ is computed using Eq. (6). The per-environment latent estimation error is then defined as

$$\hat{\epsilon}_\xi(N) := \big\|\hat{\boldsymbol{z}}_\xi^{ref} - \hat{\boldsymbol{z}}_\xi^{(N)}\big\|_2. \tag{85}$$

Aggregating over the domain-randomization ensemble $\xi \sim \mu_{env}$, an empirical distribution of $\hat{\epsilon}_\xi(N)$ is obtained. The offline encoder error function $\epsilon_{\text{sim}}(N)$ is approximated by a high quantile of this distribution:

$$\epsilon_{\text{sim}}(N) \approx \text{Quantile}_{q_\epsilon}\big(\{\hat{\epsilon}_\xi(N)\}_{\xi \sim \mu_{env}}\big), \tag{86}$$

where $q_\epsilon$ controls the conservativeness.

**Latent sensitivity for $L_c$.** Similarly, to estimate the Lipschitz constant $L_c$, the cost value functions under the reference latent and the $N$ context latent are:

$$J_c^{ref}(\xi) := J_c(\pi^\varphi|\hat{\boldsymbol{z}}_\xi^{ref}), \quad J_c^{(N)}(\xi) := J_c(\pi^\varphi|\hat{\boldsymbol{z}}_\xi^{(N)}). \tag{87}$$

The corresponding cost difference is

$$\Delta J_{c,\xi}(N) := \big|J_c^{ref}(\xi) - J_c^{(N)}(\xi)\big|. \tag{88}$$

As $\hat{\epsilon}_\xi(N) > 0$, an empirical Lipschitz ratio can be computed as

$$\hat{L}_{c,\xi}(N) := \frac{\Delta J_{c,\xi}(N)}{\hat{\epsilon}_\xi(N)}. \tag{89}$$

Collecting $\hat{L}_{c,\xi}(N)$ over environments $\xi \sim \mu_{env}$, a boot-strapping set is obtained. $L_{c,\text{sim}}(N)$ could be a conservative estimation with $q_L$ at a high quantile.

$$L_{c,\text{sim}}(N) \approx \text{Quantile}_{q_L}\big(\{\hat{L}_{c,\xi}(N)\}_{\xi \sim \mu_{env}}\big), \tag{90}$$

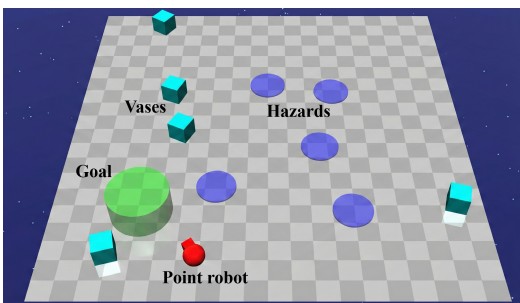

*Figure 5.* Example of POINTGOAL2 task.

**Return reduction estimation.** To investigate the relationship between $\eta$ and value function reduction terms $\Delta(\eta, \boldsymbol{z})$, similar offline calibration is conducted. For each simulated environment $\mathcal{M}_{\xi^*}$ with reference latent $\hat{\boldsymbol{z}}_\xi^{ref}$, both the risk-neutral policy $\pi_{exp}^\varphi$ and the risk-sensitive policy $\pi_\eta^\varphi$ with a given quantile parameter $\eta$ are evaluated. Then, the empirical cost reduction for environment $\mathcal{M}_{\xi^*}$ is:

$$\Delta_{c,\xi}(\eta|N) := J_c(\pi_{exp}^\varphi|\hat{\boldsymbol{z}}_\xi^{(N)}) - J_c(\pi_\eta^\varphi|\hat{\boldsymbol{z}}_\xi^{(N)}). \tag{91}$$

Aggregating over $\xi \sim \mu_{env}$, a conservative lower bound is defined:

$$\Delta_{c,sim}(\eta|N) \approx \text{Quantile}_{q_{\Delta_c}}\big(\{\Delta_{c,\xi}(\eta|N)\}_{\xi \sim \mu_{env}}\big), \tag{92}$$

where $q_{\Delta_c}$ is a low quantile.

*Remark* D.1. The conservative estimation of $\epsilon_{\text{sim}}(N)$, $L_{c,\text{sim}}(N)$ and $\Delta_{c,sim}(\eta|N)$ contributes to a safe application to the unseen real environment $\xi^*$, ensuring $\Delta_{c,sim}(\eta|N) \geq L_{c,\text{sim}}(N)\epsilon_{\text{sim}}(N)$. Intuitively, for $q_{\text{RHS}} := \min\{q_\epsilon, q_L\}$ percent random environments for the RHS and $1 - q_\Delta$ percent for the LHS, given a $N$, there will be a $\eta$ that satisfies the safety condition in Theorem 5.3. In practice, to simplify implementation without loss of generality, we approximate the expectation using the median quantile.

## E. Experimental Settings

### E.1. POINTGOAL2 Task

We evaluate our method in standard environments from the OpenAI Safety Gym suite (Ray et al., 2019), following the experimental protocol of (As et al., 2025). In particular, we consider the POINTGOAL2 task, which requires an agent to navigate to a target location while avoiding safety-critical events such as collisions with movable objects and entering hazardous regions.

As shown in Figure 5, the agent is rewarded based on progress toward the goal, measured by the reduction in Euclidean distance between successive time steps, with an additional terminal bonus upon reaching the goal. Safety costs are incurred when the agent collides with obstacles, induces excessive object motion, or enters predefined hazard zones. These reward and cost definitions are identical to those used in (Ray et al., 2019), ensuring a consistent benchmark for comparison.

***Sim2Real* gap.** To evaluate the model robustness under the *Sim2Real* gap, we introduce domain randomization over a subset of physical parameters during training and evaluation. The joint damping, actuator gear ratios, and body mass are randomized with different ranges and magnitudes in training and evaluation. Specifically, we consider three levels of OOD shifts, denoted as Mild, Medium, and High, by progressively expanding the variations of the physical parameters, as shown in Table 2.

*Remark* E.1. Compared to As et al. (2025), which employs very limited randomization during training, our method adopts substantially wider randomization intervals. This design choice is motivated by the methodological differences. As et al. (2025) incorporates an explicit pessimistic term in the cost objective, which can become overly conservative under large uncertainty sets and hinder learning. In contrast, our approach uses a latent context variable to characterize the environment,

*Table 2.* Domain randomization parameters and ranges used for training and evaluation under different OOD levels in the PointGoal2 environment. × and + denote multiplicative and additive perturbations, respectively.

| Parameter | Train | Evaluation (OOD Level) | | |
|---|---|---|---|---|
| | | **Mild** | **Medium** | **High** |
| Damping (x,y) × | [0.6, 1.0] | [0.7, 1.0] | [0.5, 1.0] | [0.3, 1.3] |
| Damping (z) × | [0.7, 1.0] | [0.8, 1.0] | [0.7, 1.0] | [0.4, 1.5] |
| Gear (x) + | [0.0, 0.2] | [0.0, 0.25] | [0.0, 0.25] | [-0.2, 0.4] |
| Gear (z) + | [0.0, 0.1] | [0.0, 0.05] | [0.0, 0.1] | [-0.2, 0.3] |
| Mass × | [0.5, 1.5] | [0.8, 2.0] | [0.5, 2.0] | [0.4, 2.0] |

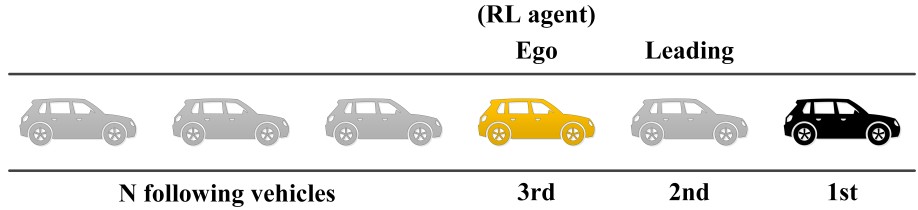

*Figure 6.* The basic setup of the vehicle platoon.

allowing us to safely expose the agent to a broader range of dynamics during training. As a result, the learned policy adapts to more diverse environments and exhibits improved robustness.

### E.2. Autonomous Driving Task for Oscillation Reduction

Traffic oscillations are a common phenomenon in traffic flow, where small perturbations introduced by a leading vehicle can be amplified and propagated upstream. Such oscillations lead to excessive fuel consumption, increased emissions, a higher risk of rear-end collisions, and reduced traffic efficiency. With the development of autonomous driving technologies, Automated Vehicles (AVs) can potentially serve as traffic regulators by adjusting their longitudinal behavior to damp these oscillations.

To study this problem, we consider a vehicle platoon scenario (Jiang et al., 2022). The scenario setup is illustrated in Figure 6. The trajectory of the first vehicle is specified using real-world trajectories extracted from EPA standard test cycles driving data. The second vehicle is a human-driven vehicle (HDV) modeled by a classical car-following model (e.g., the FVD model) (Jiang et al., 2001). The third vehicle is controlled by the proposed RL agent. The remaining $N$ following vehicles are all modeled as HDVs using the same car-following model. This setup allows us to evaluate whether the RL-controlled vehicle can mitigate the propagation of traffic oscillations while interacting with surrounding HDVs.

*Sim2Real* **gap.** All experiments are conducted in the MoJoCo platform. To capture domain shifts induced by changes in vehicle and road dynamics, we model key physics parameters including vehicle mass, aerodynamic resistances, engine torque and inertia, breaking forces, and tire-road frictions to better represent real-world system dynamics. All the physical parameters are randomized in the deployment stage to demonstrate the *Sim2Real* gap. As shown in Table 3, each parameter is initialized from its default value and then perturbed through multiplicative randomization. In particular, vehicle mass $m$ affects acceleration and braking capability; the drag coefficient $C_d$ determines the level of aerodynamic resistance, especially at higher speeds; the engine torque scaling factor $\alpha_{\text{drive}}$ is used to vary engine output; the engine inertia $I_{\text{engine}}$ controls how quickly the engine responds to throttle inputs; and the tire friction coefficient $\mu$ reflects the level of tire-road grip under different surface conditions. Together, these parameters define a practical domain randomization space for evaluating the robustness of the proposed longitudinal controller under varying vehicle and road conditions.

The settings of the state, action, reward, and cost are shown as follows.

*Table 3.* Physics randomization ranges used for training and deployment evaluation in the MuJoCo platoon environment. All values are multiplicative scale factors ($\times$) applied to the calibrated defaults: $m{=}1000\,\text{kg}$, $C_d{=}0.3$, drive force$=5000\,\text{N}$, brake force$=18000\,\text{N}$, $\tau{=}0.4\,\text{s}$, $\mu{=}2.0$.

| Parameter | Train | Deploy |
|---|---|---|
| Vehicle mass $m \times$ | $[0.9, 1.1]$ | $[0.8, 1.2]$ |
| Drag coefficient $C_d \times$ | $[0.3, 0.5]$ | $[0.3, 0.6]$ |
| Drive torque $\alpha_{\text{drive}} \times$ | $[1.5, 2.0]$ | $[1.5, 2.5]$ |
| Brake torque $\alpha_{\text{brake}} \times$ | $[0.5, 1.0]$ | $[0.2, 0.5]$ |
| Engine inertia $I_{\text{engine}} \times$ | $[1.0, 1.5]$ | $[1.0, 2.0]$ |
| Tire friction $\mu \times$ | $[0.5, 1.0]$ | $[0.3, 0.8]$ |

**State:** The real-time kinematics of the ego, leading and following vehicles are used to formulate the state representation.

$$s\left(t\right) = [v_{ego}(t), \dot{v}_{ego}(t), \ddot{v}_{ego}(t), v_{lead}(t), \dot{v}_{lead}(t), \ddot{v}_{lead}(t), l_{lead}(t), v_{follow}(t), l_{follow}(t)] \tag{93}$$

where $v_{ego}(t)$, $\dot{v}_{ego}(t)$, and $\ddot{v}_{ego}(t)$ are the speed, acceleration, and jerk of the ego vehicle, respectively. Similarly, $v_{lead}(t)$, $\dot{v}_{lead}(t)$, and $\ddot{v}_{lead}(t)$ are the variables of the leading vehicle. $l_{lead}(t)$ is the distance between the ego and leading vehicles. Similarly, $v_{follow}(t)$ and $l_{follow}(t)$ are the speed of the first following vehicle and its distance from the ego, respectively. Note that these variables can be directly measured by AV onboard sensors.

**Action:** Since we only consider the longitudinal control of the vehicle, the throttle and braking are the control actions, which could be simplified into a continuous variable $u \in [-1, 1]$. $u = 1$ represents 100% throttle and $u = -1$ means 100% braking.

**Reward:** The reward function encourages the ego vehicle to maintain a smooth and stable car-following behavior while dampening stop-and-go oscillations. The first goal is to discourage aggressive closing-in behavior when the ego vehicle is already close to the leading vehicle. Specifically, a penalty is imposed when the ego vehicle travels faster than the leading vehicle under a short spacing condition ($l(t)$ is less than a threshold $l_c$):

$$r_v\left(t\right) = -f_v\,\mathbf{1}(l(t) < l_c)\,\max(v_{ego}(t) - v_{lead}(t),\, 0)^2. \tag{94}$$

Second, to maintain efficient and reasonable car-following behavior, the ego vehicle is encouraged to maintain high speeds:

$$r_s\left(t\right) = f_s \cdot \min(v_{max}, v_{ego})^2, \tag{95}$$

where $v_{max}$ is the speed limit, and $f_v$ and $f_s$ are the weights of the components.

The third goal is to reduce speed variation and dampen the stop-and-go traffic.

$$r_a\left(t\right) = -f_a \cdot \dot{v}_{ego}(t)^2, \tag{96}$$

$$r_j\left(t\right) = -f_j \cdot \ddot{v}_{ego}(t)^2, \tag{97}$$

where $f_a$ and $f_j$ are the weights of the components.

Combining these terms, the overall vehicle agent reward is expressed as,

$$r\left(t\right) = r_v\left(t\right) + r_s\left(t\right) + r_a\left(t\right) + r_j\left(t\right). \tag{98}$$

**Cost:** Frequent speed changes of the leading vehicle may cause unsafe car following, leading to potential rear-end collision hazard. Therefore, we use inverse time-to-collision (iTTC), which is a well recognized surrogate measure for safety. Larger iTTCs indicate high risks of rear-end collisions. The iTTC between the ego vehicle and its leading vehicle can be calculated as,

$$c(t) = \max\left[\frac{\max\left(v_{ego}(t) - v_{lead}(t), 0\right)}{l_{lead}(t)}, \frac{\max\left(v_{follow}(t) - v_{ego}(t), 0\right)}{l_{follow}(t)}\right]. \tag{99}$$

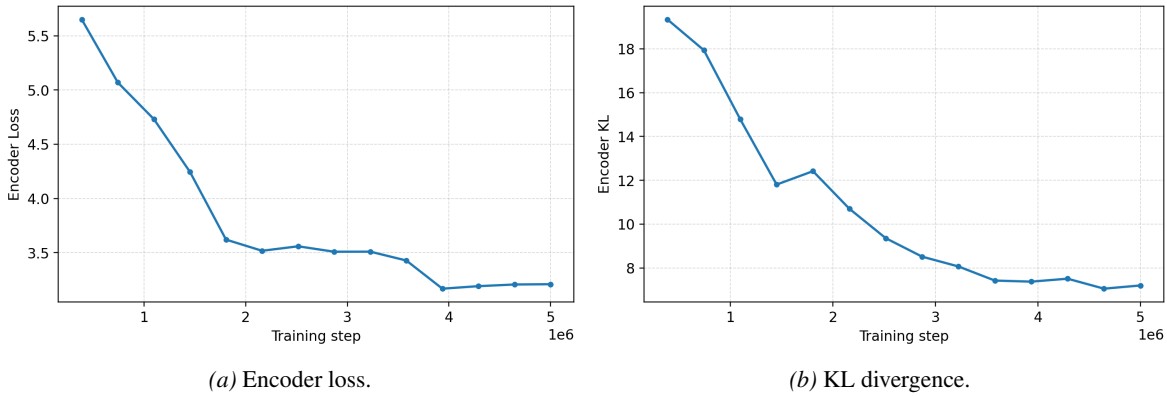

*(a)* Encoder loss.                                      *(b)* KL divergence.

*Figure 7.* Training dynamics of the encoder. (a) Encoder loss convergence. (b) KL divergence between the learned posterior and the prior during training.

*Table 4.* Component comparison of the ablation methods. R.A. denotes risk adaptation.

| Method | Backbone | Latent Encoder | Risk Adaptation |
|---|---|---|---|
| Nominal | IQN | ✗ | ✗ |
| PEARL | SAC | ✓ | ✗ |
| Ours w/o R.A. | IQN | ✓ | ✗ |
| Ours | IQN | ✓ | ✓ |

## F. Supplementary Experimental Results

### F.1. Training Stability and Encoder Behavior

Figure 7 illustrates the training dynamics of the encoder in the proposed framework for the POINTGOAL2 task. As shown in Figure 7 (a), the encoder loss consistently decreases and converges as training proceeds, indicating stable optimization and empirically supporting the convergence analysis presented in the main text. This also suggests that the encoder learns increasingly informative representations that are beneficial for downstream policy learning.

Figure 7 (b) reports the KL divergence between the encoder posterior and the prior. After an initial decrease, the KL value stabilizes in later training stages, implying that the learned posterior neither collapses to the prior nor grows unbounded. This behavior indicates that the encoder successfully captures environment-specific information while maintaining regularization, avoiding overfitting and degenerate latent representations.

Together, these results demonstrate that the encoder is well trained and provides a reliable latent context for policy learning and deployment.

### F.2. Ablation Study

An incremental ablation study is conducted from the nominal IQN backbone to the complete method for the POINTGOAL2 task. The Nominal baseline uses only the IQN backbone. Ours w/o R.A. adds the latent encoder to IQN, while Ours further introduces the risk adaptation module based on the inferred latent representation. Since risk adaptation in our framework relies on the latent context, removing the latent encoder also removes the basis for risk adaptation, making this variant equivalent to the Nominal baseline. Therefore, a separate "w/o latent encoder" baseline is not included. In addition to these internal ablations, we include PEARL (Rakelly et al., 2019) as a representative latent-context adaptation baseline with a SAC backbone.

The results are shown in Table 5. Nominal achieves high rewards but suffers from severe cost violations during deployment. Both PEARL and Ours w/o R.A. are meta-RL-style ablation methods. They exhibit more conservative behaviors and achieve substantially lower deployment costs. The IQN backbone in Ours w/o R.A. slightly outperforms the SAC backbone used in PEARL. However, the deployment costs of both baselines still exceed the safety threshold of 10. In contrast, our complete method further reduces the cost below the threshold. This benefit is also shown in Figure 3, where the blue curve

*Table 5.* Ablation results under training and deployment in POINTGOAL2 Task. R.A. denotes risk adaptation.

| | | Nominal | PEARL | Ours w/o R.A. | Ours |
|---|---|---|---|---|---|
| Train | Reward | 92.98±2.07 | 33.11±4.81 | – | 35.17±3.94 |
| | Cost | 9.70±0.92 | 9.18±0.40 | – | 9.46±0.51 |
| Deploy | Reward | 91.43±7.25 | 46.09±5.20 | 54.21±8.70 | 49.59±8.52 |
| | Cost | 29.62±3.73 | 10.75±1.91 | 10.12±1.53 | **8.86±0.85** |

*Table 6.* Action-refinement time under different hyperparameter combinations.

| $K_{\text{ref}}$ | $\alpha_r$ | $\alpha_c$ | $\beta_n$ | Time (ms) |
|---|---|---|---|---|
| 5 | 0.01 | 0.05 | 0.0 | 3.51±0.27 |
| 5 | 0.10 | 0.05 | 1.0 | 3.54±0.32 |
| 5 | 0.01 | 0.30 | 1.0 | 3.55±0.28 |
| 5 | 0.10 | 0.30 | 0.0 | 3.56±0.28 |
| 15 | 0.01 | 0.05 | 1.0 | 9.18±0.47 |
| 15 | 0.10 | 0.05 | 0.0 | 9.19±0.56 |
| 15 | 0.01 | 0.30 | 0.0 | 9.17±0.39 |
| 15 | 0.10 | 0.30 | 1.0 | 9.19±0.58 |

consistently maintains the cost below the threshold throughout deployment, demonstrating a clear safety advantage over the other baselines.

### F.3. Online Computational Overhead and Complexity Analysis

We test the running time of Algorithm 1 under different combinations of $K_{\text{ref}}$, $\alpha_r$, $\alpha_c$, and $\beta_n$ for the Autonomous Driving task. In general, the average action-refinement time is below 10 ms on a desktop equipped with an Intel Core i9-12900KF CPU and an RTX 3090 GPU, which is sufficiently fast for real-time operation under a typical 20 Hz control resolution. The online computational overhead is summarized in Table 6.

Each refinement step computes $\nabla_a Q_r$ for reward-gradient ascent and $\nabla_a Q_c$ for cost-gradient descent, both of which require forward and backward passes through the IQN critic networks. As shown in Table 6, the running time scales approximately linearly with $K_{\text{ref}}$. In contrast, the step-size parameters $\alpha_r$, $\alpha_c$, and $\beta_n$ do not affect the computation time, since they are scalar multipliers within the same JIT-compiled computational graph.

We also provide a detailed complexity analysis in terms of Big-O notation and FLOP counts for each component of the framework. The results are consistent with the empirical computation time observed in the experiments.

As shown in Table 7, let $d_s$, $d_a$, and $d_z$ denote the observation, action, and meta-latent dimensions, respectively. Let $d_\pi$, $d_Z$, and $d_\phi$ denote the hidden widths of the policy, distributional critic, and meta-encoder, respectively. Let $L_\pi$, $L_Z$, and $L_\phi$ denote their corresponding depths. In addition, $L_B$ is the number of BroNet residual blocks, $d_\tau$ is the cosine embedding dimension, $N_\tau$ is the number of quantile samples, $N_c$ is the number of critics, $M$ is the number of context transitions, and $K_{\text{ref}}$ is the number of refinement steps.

A summary of the complexity analysis is shown in Table 8. The computational complexity is dominated by action refinement, which accounts for over $90\%$ of both FLOPs and wall-clock time. The FLOP magnitudes of different components, approximately $10^5 : 10^7 : 10^9$, are generally consistent with the measured computation time, with small deviations caused by GPU parallelism and kernel-launch overhead. The cost scales linearly with $K_{\text{ref}}$, $N_\tau$, $N_c$, $L_B$, and $M$, and quadratically with $d_Z$. The full inference pipeline, from receiving a state observation to outputting a safety-refined action with $K_{\text{ref}} = 5$, takes less than 4 ms, occupying less than $8\%$ of the 50 ms control timestep. Even with $K_{\text{ref}} = 15$ refinement steps, the total inference time remains below 10 ms, corresponding to about $20\%$ utilization of the control timestep. Given the current complexity level and online computational overhead, the framework is efficient enough for real-time deployment.

Detailed complexity analysis for each component is provided below.

*Table 7.* Notation and values used in the complexity analysis.

| Symbol | Meaning | Value |
|---|---|---|
| $d_s, d_a, d_z$ | Obs./action/latent dimensions | $9, 1, 5$ |
| $d_\pi$ ($L_\pi$ layers) | Policy width (depth) | 256 (3) |
| $d_Z$ ($L_Z$ layers) | Distributional critic feature width (depth) | 512 (2) |
| $L_B$ | BroNet residual blocks per critic | 2 |
| $d_\tau$ | Cosine embedding dimension | 64 |
| $N_\tau$ | Quantile samples per forward pass | 32 |
| $N_c$ | Number of critics | 2 |
| $d_\phi$ ($L_\phi$ layers) | Meta-encoder width (depth) | 256 (3) |
| $M$ | Number of context transitions | 128 |
| $K_{\mathrm{ref}}$ | Refinement steps | 5 |

*Table 8.* Summary of computational complexity, FLOP counts, and empirical running time.

| Component | Complexity | FLOPs | Empirical time |
|---|---|---|---|
| Policy $\pi$ | $O(L_\pi d_\pi^2)$ | $\sim 10^5$ | 0.03 ms |
| Meta-encoder $q_\phi$ | $O(M L_\phi d_\phi^2)$ | $\sim 10^7$ | 0.08 ms |
| $\eta$ computation | $O(1)$ | negligible | 0.14 ms$^\dagger$ |
| Action refinement | $O(K_{\mathrm{ref}} N_c N_\tau L_B d_Z^2)$ | $\sim 10^9$ | 3.51 ms |

$^\dagger$ The $\eta$ computation has negligible FLOPs; the measured 0.14 ms mainly reflects fixed GPU kernel-launch overhead rather than arithmetic computation.

**Policy $\pi(s, z)$.** The policy is a three-layer MLP with input dimension $d_s + d_z$. Its computational cost is

$$C_\pi = (d_s + d_z)d_\pi + (L_\pi - 1)d_\pi^2 + d_\pi d_a = O(L_\pi d_\pi^2) \approx 1.3 \times 10^5 \text{ FLOPs.} \tag{100}$$

**Meta-encoder $q_\phi(s, s', a, r, c)$.** The meta-encoder is a three-layer MLP applied independently to each of $M$ context transitions. The input dimension is $2d_s + d_a + 2 = 21$. The resulting embeddings are then aggregated using Product-of-Gaussians, whose cost is $O(Md_z)$ and is negligible. The computational cost is

$$C_\phi = M\left[(2d_s + d_a + 2)d_\phi + (L_\phi - 1)d_\phi^2 + 2d_\phi d_z\right] = O(M L_\phi d_\phi^2) \approx 1.8 \times 10^7 \text{ FLOPs.} \tag{101}$$

**Distributional critic forward pass.** The feature extractor maps $(s, a, z)$ to $\mathbb{R}^{d_Z}$. Then, $N_\tau$ quantile levels are sampled, embedded via cosine basis functions, projected to $\mathbb{R}^{d_Z}$, and element-wise multiplied with the hidden state. Finally, $N_c$ BroNet critics, each with $1 + 2L_B$ dense layers of width $d_Z$ plus a scalar head, produce the final quantile values. The computational cost is

$$C_Z = L_Z d_Z^2 + N_\tau d_\tau d_Z + N_c N_\tau (1 + 2L_B)d_Z^2 = O(N_c N_\tau L_B d_Z^2) \approx 8.4 \times 10^7 \text{ FLOPs.} \tag{102}$$

**Action refinement.** Each refinement step requires three operations through the distributional critics: (i) one $Z_c$ forward pass to evaluate $Q_c^\eta$ for early stopping, (ii) one $Z_r$ forward and backward pass to compute $\nabla_a Q_r$, and (iii) one $Z_c$ forward and backward pass to compute $\nabla_a Q_c^\eta$. This gives 3 forward passes and 2 backward passes per step. Approximating the backward pass as having the same cost as the forward pass, each refinement step costs $5C_Z$. Therefore,

$$C_{\mathrm{ref}} = 5K_{\mathrm{ref}} C_Z = O(K_{\mathrm{ref}} N_c N_\tau L_B d_Z^2) \approx 2.1 \times 10^9 \text{ FLOPs.} \tag{103}$$

### F.4. Video

We rendered videos for different methods, which could further illustrate the superiority of our method. The video could be found in https://youtu.be/Mn9avTYnMa8.

