# OpenReview forum: "Transferable Reinforcement Learning via Probabilistic Latent Embeddings and Dynamic Policy Adaptation for Sim-to-Real Deployment"
_ICML.cc/2026/Conference — ICML 2026 regular_

### Official Review · Reviewer_xxS7 · 2026-03-01

**Soundness:** 2
**Presentation:** 3
**Significance:** 2
**Originality:** 3
**Overall Recommendation:** 4
**Confidence:** 4

**Summary:**

This paper introduces a transferable RL for simulation-to-real world scenarios. The problem is formulated as constrained Markov decision process (CMDP) over dynamic environments, The proposed framework mainly focus on combining distributional RL with latent context adaptation to achieve conservatism over unseen scenarios. The authors evaluate their method on the well-established benchmark OpenAI Safety Gym.

**Compliance With Llm Reviewing Policy:**

Affirmed.

**Final Justification:**

After considering the paper and the rebuttal to my review and to other reviewers, I raised the score to 4 as the authors addressed my main concerns.

**Key Questions For Authors:**

1. What is the complexity of the proposed framework compared to the baselines? Is it realistic for real world scenarios to design highly complicated frameworks?

2. How your proposed method be formulated in offline RL context?

3. How will the proposed framework perform in real world scenario beyond simulations and enviroments changing?

**Limitations:**

In my opinion, a limitation section (or a paragraph in the conclusion) is needed to highlight the potential limitations of their work.

**Strengths And Weaknesses:**

Strengths:

1. The paper provides solid theoretical coverage with clear and well-structured proofs.

2. The problem of the distributional shift between simulation and real world scenarios is a very important and timely problem in the RL domain.

3. The organization, writing quality, and overall clarity of the paper are very good, and the presentation is easy to follow.

4. The integration of distributional RL and meta-RL is clearly motivated and well justified within the proposed framework.

Weaknesses:

1. The main drawback in the paper is the experiments. The authors, very well, motivated the need for models that address the gap between simulation and real world scenarios. However, they only include simulation-based environments, even if they change the seed and the dynamics of the scenarios.

2. The work lacks comparing strong baselines from the meta-RL domain (Pearl for example) and distributional RL.

3. The proposed framework can be very complicated. A complexity analysis is missing.

4. An ablation study is not presented to highlight the significance of each module in the framework.

5. some writing issues appear, for example punctuation in the equations.

6. While the authors focus on real world issues, it has been proven that online RL does not suit real world deployments due to safety concerns.

---

> ### Author Rebuttal · Authors · 2026-03-31
>
> Thank you so much for your valuable feedback.
>
> [Weakness 1] We provide a new case study regarding autonomous driving systems. Specifically a cooperative longitudinal control task, is added as additional experiments. In this task, an autonomous vehicle is controlled by the RL agent to reduce traffic oscillations. Moreover, in the new experiment, we model key physics parameters including vehicle mass, aerodynamic resistances, engine torque and inertia, breaking forces, and tire-road frictions to better represent real-world system dynamics. All the physical parameters are randomized in the deployment stage to demonstrate the Sim2Real gap.
> We hope the new case study could better illustrate the practical applicability of our method. More details regarding this case study can be found in the reponses to "*Reviewer bPoU* [Question 2]".
> Besides, we indeed plan to conduct real-world implementation using a testing AV. As the rebuttal time is limited, we will put it in our future work in the revised manuscript.
>
> [Weakness 2] A PEARL-style baseline is actually included in the comparison, which is denoted as "Ours w/o Risk Adaptation" shown as orange curves in Figure 3. Different from the original PEARL, the "Ours w/o Risk Adaptation" baseline uses distributional RL algorithm for fair comparison with the proposed method. Nonetheless, we further add the original PEARL method as another baseline in the ablation study. We refer you to the "*Reviewer PHwa* [Question 1]" for more details.
>
> [Weakness 3] Analyzing the complexity of the modeling framework mathematically is tricky. To evaluate the computational load and validate the feasibility of real-time implementation, we analyze the computational overhead. We refer you to the "*Reviewer hReh* [Question 3]" for more details.
>
> [Weakness 4] The ablation studies is added. Please see our responses to your "[Weakness 2]".
>
> [Weakness 5] We have revised the usage of punctuation in equations to enhance the rigor of writing. We will also carefully proofread the entire manuscript to correct other potential typos and errors in the camera-ready version.
>
> [Weakness 6] It is true that the exploration of online RL may cause unsafe actions, and this is the exact motivation of our proposed model: using distributional RL to allow risk levels of the policy can be adjusted dynamically at the deployment stage, based on the estimation accuracy of the latent context variable. As shown in the experimental results, our model can achieve relatively high performance (blue line in Figure 3(b)) while keeping cost always under the threshold (blue line in Figure 3(a)). However, in real world deployment, it is true that numerous factors may have impacts on safety. As a result, for safety-critical systems such as autonomous driving, a safety check module (e.g., a safety envelope) is usually added on top of the RL agent's actions. However, with the proposed approach, the probability of triggering the intervention of the safety check module would be greatly reduced.
>
> [Question 1] Please see our responses to your "[Weakness 3]".
>
> [Question 2] The proposed framework can be implemented in offline RL context using trajectories from real world environments. However, because collecting real-world trajectories is costly, the amount of available data is often limited and biased. As a result, both the encoder of the context variable $z$ and the estimates of the value function distribution $Z_{r}$, $Z_{c}$ may be inaccurate. In particular, limited coverage of system dynamics can lead to poor estimation of the context variable, while the lack of edge-case samples can result in inaccurate estimation of the tails of the value function distributions. Nevertheless, we believe that offline RL is a promising direction for safety-critical applications, and we plan to explore this in future work.
>
> [Question 3] The proposed framework is applicable to real-world tasks such as autonomous driving, as discussed in the responses to "[Weakness 1]". In the current work, we mainly consider adaptation to an unknown but fixed environment during deployment. Handling continuously changing environments is an interesting direction for future work. One possible direction is to design a detection mechanism for the changes in the environment, such as a sudden shift from a dry pavement to a icy pavement. By integrating the detection module with our framework, the new method could better handle dynamically changing environments.
>
> [Limitation] We add a limitation paragraph to discuss the challenge of changing environments, which is shown as follows.
>
> "The current framework assumes that the deployment environment is unknown but relatively fixed. Therefore, handling abrupt or continuously changing environments remains an important direction for future work."

---

> > ### Author Rebuttal · Reviewer_xxS7 · 2026-04-02
> >
> > Thanks for the authors feedback.
> >
> > For weakness 3, can the authors elaborate more how evaluating the complexity analysis mathematically is tricky?

---

> > > ### Author Response · Authors · 2026-04-02
> > >
> > > April 7 update:
> > >
> > > We are wondering whether our responses regarding the complexity analysis have addressed your  remaining concern and please let us know if you have further questions. Thank you for your continuous consideration of our work!
> > >
> > > ---
> > >
> > > April 2 response:
> > >
> > > Thank you for your valuable feedback and the constructive follow-up comments.
> > >
> > > To address your question more directly, we now provide detailed complexity analysis in terms of Big-O and FLOP counts for each component of the framework. Results show that they are consistent with the computation time observed in the experiment.
> > >
> > > Let $d_s$, $d_a$, $d_z$ denote the observation, action, and meta-latent dimensions; $d_\pi$, $d_Z$, $d_\phi$ denote the hidden widths of the policy, distributional critic, and meta-encoder; $L_\pi$, $L_Z$, $L_\phi$ are their respective depths; $L_B$ is the number of BroNet residual blocks; $d_\tau$ is the cosine embedding dimension; $N_\tau$ is the number of quantile samples; $N_c$ is the number of critics; $M$ is the number of context transitions; and $K_{ref}$ is the number of refinement steps.
> > >
> > > |Symbol|Meaning|Value|
> > > |---|---|:---:|
> > > |$d_s,d_a,d_z$|obs/action/latent dim|9,1,5|
> > > |$d_\pi$ ($L_\pi$ layers)|policy width (depth)|256 (3)|
> > > |$d_Z$ ($L_Z$ layers)|distributional critic feature extractor width (depth)|512 (2)|
> > > |$L_B$|BroNet residual blocks per critic|2|
> > > |$d_\tau$|cosine embedding dim|64|
> > > |$N_\tau$|quantile samples per forward pass|32|
> > > |$N_c$|number of critics|2|
> > > |$d_\phi$ ($L_\phi$ layers)|meta-encoder width (depth)|256 (3)|
> > > |$M$|number of context transitions|128|
> > > |$K_{ref}$|refinement steps|5|
> > >
> > > A summary of the complexity analysis is shown in the following table:
> > >
> > > |Component|Complexity|FLOPs|Empirical time|
> > > |---|---|:---:|:---:|
> > > |Policy $\pi$|$O(L_\pi\cdot d_\pi^2)$|$\sim10^5$|0.03 ms|
> > > |Meta-encoder $q_{\phi}$|$O(M\cdot L_\phi\cdot d_\phi^2)$|$\sim10^7$|0.08 ms|
> > > |$\eta$ computation|$O(1)$|negligible|0.14 ms$^\dagger$|
> > > |Action refinement|$O(K_{ref}\cdot N_c\cdot N_\tau\cdot L_B\cdot d_Z^2)$|$\sim10^9$|3.51 ms|
> > >
> > > $^\dagger$ The $\eta$ computation has negligible FLOPs; the measured 0.14 ms reflects fixed GPU kernel launch costs rather than arithmetic.
> > >
> > > Overall, the complexity is dominated by the action refinement, which accounts for over 90\% of the FLOPs and the wall-clock time. The FLOP magnitudes of different components ($10^5:10^7:10^9$) generally match the measured computation time, with small deviations caused by GPU parallelism and kernel launch overhead. The cost scales linearly with $K_{ref}$, $N_\tau$, $N_c$, $L_B$, and $M$, and quadratically with $d_Z$. Given the current complexity level, the framework is efficient enough for real-time deployment.
> > >
> > > Detailed complexity analysis for each component is shown as follows.
> > >
> > > - Policy $\pi(s, z)$: Three-layer MLP with input $(d_s+d_z)$.
> > >
> > > $C_\pi=(d_s+d_z)d_\pi+(L_\pi-1)d_\pi^2+d_\pi d_a=O(L_\pi d_\pi^2)\approx1.3\times10^5\text{ FLOPs}$
> > >
> > > - Meta-encoder $q_{\phi}(s, s', a, r, c)$: Three-layer MLP applied independently to each of $M$ context transitions (input dimension: $2d_s + d_a + 2 = 21$), followed by Product-of-Gaussians aggregation ($O(M \cdot d_z)$, negligible).
> > >
> > > $C_\phi=M\left[(2d_s+d_a+2)d_\phi+(L_\phi-1)d_\phi^2+2d_\phi d_z\right]=O(ML_\phi d_\phi^2)\approx1.8\times10^7\text{ FLOPs}$
> > >
> > > - Distributional critic forward pass: The feature extractor maps $(s, a, z)$ to $\mathbb{R}^{d_Z}$; $N_\tau$ quantile levels are sampled, embedded via cosine basis functions, projected to $\mathbb{R}^{d_Z}$, and element-wise multiplied with the hidden state; $N_c$ BroNet critics (each with $1 + 2L_B$ dense layers of width $d_Z$ plus a scalar head) produce the final quantile values.
> > >
> > > $C_Z=L_Zd_Z^2+N_\tau d_\tau d_Z+N_cN_\tau(1+2L_B)d_Z^2=O(N_cN_\tau L_Bd_Z^2)\approx8.4\times10^7\text{ FLOPs}$
> > >
> > > - Action refinement: Each step requires three operations through the distributional critics: (i) one $Z_c$ forward pass to evaluate $Q_c^\eta$ for early stopping, (ii) one $Z_r$ forward + backward pass to compute $\nabla_a Q_r$, and (iii) one $Z_c$ forward + backward pass to compute $\nabla_a Q_c^\eta$. This totals 3 forward + 2 backward passes per step; approximating backward $\approx$ forward in FLOPs gives $5 C_Z$ per step.
> > >
> > > $C_{ref}=5K_{ref}C_Z=O(K_{ref}N_cN_\tau L_Bd_Z^2)\approx2.1\times10^9\text{ FLOPs}$

---

### Official Review · Reviewer_PHwa · 2026-03-13

**Soundness:** 3
**Presentation:** 2
**Significance:** 2
**Originality:** 2
**Overall Recommendation:** 4
**Confidence:** 2

**Summary:**

This paper proposes a Policy Adaptation for Sim-to-Real Deployment.  First, they learn a  policy with latent context in the simulated environments. When deploying this policy, it utilizes a small number of real interactions to infer the specific context of the current environment. Then it also dynamically adjusts its risk sensitivity based on the uncertainty of this inference, so that the policy is initially conservative, then progressively more permissive.

**Compliance With Llm Reviewing Policy:**

Affirmed.

**Final Justification:**

Most of my concern are resolved. Therefore, I raise my score to 4.

**Key Questions For Authors:**

1. Are there any ablation studies to verify that each component is necessary? It seems to me that you have too many components. I was wondering how each component is contributing to the overall performance.

2, I was wondering whether the latent z can really learn the dynamics information or capture the dynamics difference.

**Limitations:**

See question and weakness.

**Strengths And Weaknesses:**

The method seems to be well justified, supported by the theoretical analysis. Each component is well motivated.  They studied an important cross-domain generalization problem, while they also have to ensure the policy is safe in the policy deployment, which I think is a hard problem. They also provide additional analysis to ensure safety. Empirical results validate their method, receiving lower cost but higher reward compared with the baseline.


Weakness:

The method looks too complicated to me, and the paper is a little bit hard to follow. And the complex method might be hard to tune in the real world practive.

---

> ### Author Rebuttal · Authors · 2026-03-31
>
> Thank you so much for your valuable feedback.
>
> [Weakness] The key idea of the proposed framework consists of two core components: 1) inferring the hidden real-world environment from a small amount of interaction data using a latent context variable; and 2) dynamically adjusting the deployed policy to be risk-sensitive according to how confident this inference is. When the environment estimate is uncertain at the beginning of the deployment, the policy behaves more conservatively to ensure safety. When the estimate becomes more accurate, the policy becomes less conservative to improve performance. The proposed method is not difficult to tune in real-world applications. To show that the proposed method is easily tunable in real world applications, we added an additional case study on autonomous driving, namely a cooperative longitudinal control task. This experiment further shows that the method is applicable in realistic settings. We refer you to the response to "*Reviewer bPoU* [Question 2]" for more details.
>
> [Question 1] An ablation study is added. Specifically, we perform an incremental ablation study from the nominal IQN backbone to the complete method. The "Nominal" baseline uses only the IQN backbone. The "Ours w/o Risk Adaptation" baseline adds the latent encoder to IQN, and the "Ours" model further introduces the risk adaptation module based on the inferred latent representation. Since risk adaptation in our framework relies on the latent context, removing the latent encoder also removes the basis for risk adaptation, making this variant equivalent to the "Nominal" baseline. Therefore, a separate "w/o latent encoder" baseline is not included. In addition to these internal ablations, we also include PEARL as a representative latent-context adaptation baseline with the SAC backbone.
>
> |Method|Backbone|Latent Encoder|Risk Adaptation|
> |---|:---:|:---:|:---:|
> |Nominal|IQN|N|N|
> |PEARL|SAC|Y|N|
> |Ours w/o R.A.|IQN|Y|N|
> |Ours|IQN|Y|Y|
>
> The results are shown as follows. "Nominal" achieves high rewards but suffer from severe cost violations. Both "PEARL" and "Ours w/o Risk Adaptation" are meta RL-style ablation methods, which exhibit conservative behaviors and have substantially lower costs during deployment. The IQN backbone is slightly better than the SAC backbone in PEARL. But note that the costs of both baselines still exceeds the threshold (i.e., 10). Our method further lowers the cost, as shown in the following Table. The benefit of our method can also be seen in Figure 3, where the blue curve consistently maintains the cost below the threshold throughout the entire deployment process, demonstrating a clear safety advantage over other baselines.
>
> |||Nominal|PEARL|Ours w/o R.A.|Ours|
> |---|---|---|---|---|---|
> |Train|reward|92.98±2.07|33.11±4.81|-|35.17±3.94|
> ||cost|9.70±0.92|9.18±0.40|-|9.46±0.51|
> |Deploy|reward|91.43±7.25|46.09±5.20|54.21±8.70|49.59±8.52|
> ||cost|29.62±3.73|10.75±1.91|10.12±1.53|8.86±0.85|
>
> [Question 2] In our framework, $z$ is trained from transitions $(s,a,r,c,s')$ collected from different environments in simulation. Since different environments produce transitions with different patterns, the latent encoder is designed to extract these environment-specific features from the transition data. This idea has also been validated in previous meta-RL works such as PEARL. In our work, the effectiveness of $z$ is further supported by the ablation study results: the cost metrics of meta RL-style methods are significantly lower than the "Nominal" baseline, suggesting that $z$ indeed captures useful environnment-related information and improve safety.

---

> > ### Author Rebuttal · Reviewer_PHwa · 2026-04-03
> >
> > Thanks for the response. Based on my limited knowledge in this domain, my questions are basically resolved. For now, I will keep my score, but I might adjust it based on the authors' discussion with other reviewers.

---

> > > ### Author Response · Authors · 2026-04-03
> > >
> > > April 7 update:
> > >
> > > Thank you again for your thoughtful follow-up. Since you mentioned that your final assessment may depend on the broader discussion, we would like to briefly summarize the current status and the main revisions made in response to the reviews.
> > >
> > > To the best of our understanding, the main questions raised across the reviews have now been addressed through the rebuttal and follow-up discussion. In particular, Reviewer hReh indicated that our response had addressed their concerns and suggested that they would raise their score. Reviewer bPoU gave the paper a supportive score, and their original concerns largely overlap with issues that have now been carefully clarified in the rebuttal and discussion. Reviewer xxS7 raised additional questions during the discussion period, and we responded to them in detail. Overall, we hope the broader discussion record provides a clearer picture of both the paper and the revisions we have made.
> > >
> > > More specifically, the rebuttal and discussion have led to the following improvements:
> > >
> > > - A new experiment has been added to demonstrate how the proposed framework can be applied to autonomous driving systems. The superior performance over the baselines further demonstrates the contribution of this work.
> > > - An ablation study together with an additional baseline has been added to illustrate the effectiveness of each component of the proposed method.
> > > - The online computational overhead and complexity analysis have been analyzed to demonstrate the efficiency and feasibility of the proposed method for real-time implementation.
> > > - Further discussion has been added on the potential applications and limitations of the proposed method.
> > > - Several minor typos have been corrected to improve the rigor of the paper.
> > >
> > > While not all reviewers have posted final follow-up messages yet, we hope that the broader discussion and these concrete revisions are helpful for your final assessment. As the final justification deadline is approaching, we are wondering whether the reviewer could adjust the score. We sincerely appreciate your consideration of our work!
> > >
> > > ---
> > >
> > > April 3 response:
> > >
> > > Thank you for your positive feedback. We are glad that our response resolved your questions. We completely understand your decision to follow the discussions with other reviewers, and we appreciate your continuous consideration of our work.

---

### Official Review · Reviewer_hReh · 2026-03-13

**Soundness:** 3
**Presentation:** 3
**Significance:** 3
**Originality:** 3
**Overall Recommendation:** 4
**Confidence:** 4

**Summary:**

This paper studies safe Sim2Real transfer for CMDP-based reinforcement learning, where policies trained in simulation may violate safety constraints after deployment. The method introduces a latent context variable inferred by an encoder $q_\phi(z\mid D_\xi)$ from transition context sets. Policy/value learning is built on IQN with reward and cost critics plus a Lagrangian constraint mechanism.

At deployment, the method uses dynamic risk regulation: upper-tail cost quantiles indexed by $\eta$, plus projected action refinement. The paper claims theory for latent-conditioned Bellman contraction, stability of alternating updates, and safety conditions based on latent estimation error. Experiments shows consistent better performances in terms of both cost and reward, as compared with Nominal, Domain Randomization, SPiDR, and Robust RL. Overall the contribution is solid, but further justification is needed.

**Compliance With Llm Reviewing Policy:**

Affirmed.

**Final Justification:**

This paper studies safe Sim2Real transfer for CMDP-based reinforcement learning, where policies trained in simulation may violate safety constraints after deployment. The method introduces a latent context variable inferred by an encoder $q_\phi(z\mid D_\xi)$ from transition context sets. Policy/value learning is built on IQN with reward and cost critics plus a Lagrangian constraint mechanism.

At deployment, the method uses dynamic risk regulation: upper-tail cost quantiles indexed by $\eta$, plus projected action refinement. The paper claims theory for latent-conditioned Bellman contraction, stability of alternating updates, and safety conditions based on latent estimation error. Experiments shows consistent better performances in terms of both cost and reward, as compared with Nominal, Domain Randomization, SPiDR, and Robust RL. Overall the contribution is solid, and the authors provide sufficient justification of the assumptions and the results during rebuttal phase.

**Key Questions For Authors:**

1. Can you provide guarantee if $\xi^\star$ is outside the training support? Additionally, if $\mu_{env}(\xi^\star)$ is really small and training is conducted with finite interactions, how would the performance of your algorithm degrade?
2. Intuitively, this latent-context-encoder-based approach allows the agent to identify the environment first, then execute the optimal policy for this environment. Why don't people just use a history-dependent policy so that the algorithm can automatically identify the environment?
3. What is the online overhead of Algorithm 1 (e.g., $K_{ref}$, gradient evaluations per step), and how sensitive are outcomes to $(\alpha_r, \alpha_c, \beta_n)$?

**Limitations:**

see weaknesses

**Strengths And Weaknesses:**

## Strengths

1. Appendix C provides an explicit projected action-refinement procedure (Algorithm 1), making the risk-sensitive policy operational.

2. The theoretical guarantee is comprehensive: The paper includes contraction/fixed-point arguments for latent-conditioned Bellman operators (Lemma 4.1, Theorem 4.2) and a stability claim for alternating optimization (Theorem 4.4).

3. The empirical performance is strong, and aligns with the theory: the paper reports deployment trajectories under dynamic adaptation (Fig. 3) and OOD sensitivity (Fig. 4), aligning with the Sim2Real claim.

## Weaknesses
1. The theoretical guarantee highly depends on the assumption that the true environment is covered by the training distribution. Since this is a core assumption, I would suggest put it the main text instead of in the appendix.


2. Typo: Appendix A.10 is titled “Proof of Theorem 5.5”. I guess the authors are referring to Theorem 5.3?

3. Evaluation is limited to POINTGOAL2, with no additional Safety Gym tasks, broader continuous-control benchmarks, or real-system transfer.

4. Although Sec. 2 discusses latent-context/meta-RL methods, experiments miss baselines like direct PEARL/VariBAD-style or comparable test-time adaptation baselines.

5. Table 1/Figs. 3–4 do not show uncertainty intervals despite multiple seeds, so statistical reliability is unclear.

---

> ### Author Rebuttal · Authors · 2026-03-31
>
> Thank you so much for your valuable feedback.
>
> [Weakness 1] We will move this core assumption to the main text in the camera-ready version.
>
> [Weakness 2] The typo is revised. We will also carefully proofread the entire manuscript to correct other potential typos and errors in the camera-ready version.
>
> [Weakness 3] An autonomous driving scenario, specifically a cooperative longitudinal control task, is added as additional experiments. In this task, an Automated Vehicle (AV) is controlled by the RL agent to reduce traffic oscillations and improve platoon stability. We refer you to the response to "*Reviewer bPoU* [Question 2]" for more details.
>
> [Weakness 4] A PEARL-style baseline is actually included in the comparison, which is denoted as "Ours w/o Risk Adaptation" shown in Figure 3 as orange curves. Different from the original PEARL, the "Ours w/o Risk Adaptation" baseline uses distributional RL algorithm for fair comparison with the proposed method. Nonetheless, we further add the original PEARL method as another baseline as you suggested in the ablation study.  We refer you to the response to "*Reviewer PHwa* [Question 1]" for more details.
>
> [Weakness 5] Uncertainty intervals (mean±std) have been added to Table 1 and Figs. 3–4. Across methods, the standard deviations are generally moderate, but the baselines tend to exhibit noticeably larger variance, especially in deployment. By contrast, our method shows comparatively stable performance, while consistently achieving the lowest cost.
>
> [Question 1] As stated by the assumption, $\xi^{\*}$ does not necessarily the same as the environments in the training process. However, the deviation between the simulation environment and the real environment could not be too far. $\xi^{\*}$ should falls within the domain randomization distribution $\mu_{env}$. If $\mu_{env}$ is very small and training is conducted with limited interactions, the performance will indeed degrade. However, our main contribution is not to address limitations in the training stage, but to improve safety and adaptability during deployment with limited real-world interaction. In practice, collecting interactions in simulators is relatively cheap, so the policy can be trained sufficiently before deployment.
>
> [Question 2]  A purely history-dependent policy may be impractical in real applications because the environment is highly complex and cannot be reliably identified from historical data alone. In autonomous driving systems, for example, engine/brake response and tire-road interaction may vary in many possible combinations. Even if policies were trained under different historical patterns, the agent would still struggle to infer which underlying environment the current situation belongs to. In our method, a latent context variable is introduced to capture such hidden environment characteristics, especially those that are difficult to describe explicitly in physical terms or never appeared before.
>
> [Question 3]  The running time of Alg. 1 with different combination of $K_{ref}$, $\alpha_r$, $\alpha_c$, $\beta_n$ is tested. In general, the average action refinement time is under $10$ ms on an desktop with Intel Core i9-12900KF (12th Gen) CPU and RTX 3090 GPU, which is fast enough for real-time operation with a typical 20 Hz control resolution. The online computational overhead is summarized the following Table.
>
> |$K_\text{ref}$|$\alpha_r$|$\alpha_c$|$\beta_n$|Time(ms)|
> |:---:|:---:|:---:|:---:|:---:|
> |5|0.01|0.05|0.0|3.51±0.27|
> |5|0.10|0.05|1.0|3.54±0.32|
> |5|0.01|0.30|1.0|3.55±0.28|
> |5|0.10|0.30|0.0|3.56±0.28|
> |15|0.01|0.05|1.0|9.18±0.47|
> |15|0.10|0.05|0.0|9.19±0.56|
> |15|0.01|0.30|0.0|9.17±0.39|
> |15|0.10|0.30|1.0|9.19±0.58|
>
> Each refinement step computes $\nabla_a Q_r$ (reward gradient ascent) and $\nabla_a Q_c$ (cost gradient descent), both requiring forward + backward passes through the IQN critic networks. As shown in the table, timing scales linearly with $K_\text{ref}$. Step sizes ($\alpha_r$, $\alpha_c$, $\beta_n$) do not affect computation time, as they are scalar multipliers within the same JIT-compiled compute graph.
>
> We further analyze the online computational overhead breakdown as shown in the table below. The full inference pipeline, from receiving a state observation to outputting a safety-refined action ($K_\text{ref}=5$), takes less than 4 ms, occupying only less than 8\% of the 50 ms control timestep. Action refinement dominates the overhead, while policy evaluation, meta-adaptation, and safe deployment $\eta$ computation are negligible. Even with $K_\text{ref} = 15$ refinement steps, the total inference time remains under 10 ms (20\% utilization), confirming that the method is fast enough for real-time deployment.
>
> |Component|Time(ms)|
> |---|:---:|
> |Policy forward pass|0.03±0.02|
> |$z$ inference|0.08±0.02|
> |$\eta$ computation|0.14±0.03|
> |Action refinement|3.51±0.27|

---

> > ### Author Rebuttal · Reviewer_hReh · 2026-04-03
> >
> > The rebuttal addressed my concerns. I will raise my score accordingly.

---

> > > ### Author Response · Authors · 2026-04-03
> > >
> > > April 7 update:
> > >
> > > As the final justification deadline is approaching, we are wondering whether the reviewer could adjust the score. Thank you very much for your consideration!
> > >
> > > ---
> > >
> > > April 3 response:
> > >
> > > Thank you for your positive feedback. We are glad our rebuttal addressed your concerns and greatly appreciate your decision to raise the score accordingly.

---

### Official Review · Reviewer_bPoU · 2026-03-14

**Soundness:** 3
**Presentation:** 4
**Significance:** 4
**Originality:** 4
**Overall Recommendation:** 5
**Confidence:** 4

**Summary:**

The paper provides a novel reinforcement learning framework to deal with the challenging issue, which usually happens when a deep reinforcement learning agent trained in simulators was deployed in the real environment. It enables safe and efficient policy transfer via probabilistic latent embedding and dynamic policy adaptation. Furthermore, it incorporates a distributional RL formulation to promote safety and improve efficiency under the Sim2Real gap.

**Compliance With Llm Reviewing Policy:**

Affirmed.

**Key Questions For Authors:**

1) If a specific application to be selected to demonstrate the implementation of framework proposed in the paper, which will be the choice of authors?  Why?
2) Could it be possible to give a brief description in the paper for the usage of framework in the domain of autonomous driving?

**Limitations:**

Besides the description of framework, more discussion concerning the specific application in real world could be appreciated.

**Strengths And Weaknesses:**

The strength of paper is that a novel reinforcement learning framework is introduced to deal with the challenging issue, which usually happens when a deep reinforcement learning agent trained in simulators was deployed in the real environment.
How to deal more practically with the trade-off between the safety and performance during real-world deployment becomes the weakness of paper.

---

> ### Author Rebuttal · Authors · 2026-03-31
>
> Thank you so much for your valuable feedback.
>
> [Weakness] The trade-off between safety and performance during real-world deployment inherently exists in many safety-critical cyber-physical systems such as autonomous driving. It is a common practice to prioritize safety over other objectives, which also aligns with the main design principle of our method. It dynamically adjusts the risk-sensitivity parameter during deployment, maintaining safety while improving performance. In the POINTGOAL2 task, our model can achieve relatively high performance (blue line in Figure 3(b)) while keeping cost always under the threshold (blue line in Figure 3(a)).
>
> [Question 1] We would apply the proposed framework in the autonomous driving system because 1) safety is the highest priority, and our risk-sensitive policy enables the agent to behave safely, even facing Sim2Real mismatch; 2) testing autonomous vehicles (AV) in real-world scenarios is both expensive and associated with public safety risks, and our method provides a practical solution by training the driving policies in simulation and enabling seamless transfer to real-world.
>
> [Question 2] We provide a new case study regarding autonomous driving systems. Below we discribe the problem setting: Oscillations are common in traffic flow, where small perturbations introduced by a leading vehicle can be amplified and propagated upstream. Such oscillations lead to excessive fuel consumption and a higher risk of rear-end collisions. AVs can potentially serve as traffic regulators by adjusting their longitudinal speeds to damp these oscillations. We mimic the settings in [1], where a AV is controlled by an RL agent to reduce traffic oscillations.
>
> The kinematics of the AV, leading and following vehicles are used to formulate the state representation. The throttle and braking are the control actions. The reward function encourages the AV to maintain a stable car-following behavior while dampening traffic oscillations. The cost is calculated by inverse time-to-collision (iTTC) and the budget is set to 20. The Sim2Real gap is introduced by changes in vehicle and road dynamics including engine/brake response and tire-road interactions. The experiments are conducted in the MoJoCo platform. Experimental results show that the control policy can successfully dampen oscillations while ensuring safety, even transferring to OOD scenarios. Detailed settings and results would be added in the camera-ready version. Key results are summarized below.
>
> Besides reward and cost, we also report: oscillation ratio (O.R.), jerk, and fuel consumption. The O.R. is defined as the ratio of the AV’s speed standard deviation to that of the preceding HDV over an episode, where a lower value indicates a greater capability of damping oscillations. If the default controller (i.e., the FVD car-following model) is applied, the metrics are O.R. (0.922±0.017), Jerk (12.329±0.195), and Fuel (0.440±0.008). The results of different RL-based controllers are summarized below.
>
> ||Metric|Ours|Ours w/o R.A.|Nominal|D.R.|SPiDR|Robust|
> |---|---|---|---|---|---|---|---|
> |Train|Rew.|434.10±67.06|-|477.70±55.44|455.19±67.43|441.34±75.59|434.81±82.98|
> ||Cost|14.00±1.27|-|19.27±0.94|15.38±1.41|15.55±1.88|21.03±1.72|
> ||O.R.|0.43±0.13|-|0.45±0.12|0.61±0.133|0.45±0.09|0.50±0.10|
> ||Jerk|2.96±1.77|-|3.23±0.62|4.85±1.65|3.45±0.72|4.06±1.98|
> ||Fuel|0.316±0.029|-|0.311±0.005|0.341±0.131|0.316±0.006|0.324±0.074|
> |Deploy|Rew.|364.85±160.74|396.59±175.33|284.44±275.72|405.28±175.50|279.88±200.05|365.99±181.04|
> ||Cost|16.80±3.33|19.31±4.31|30.98±3.99|21.58±4.47|18.53±4.15|27.80±4.00|
> ||O.R.|0.47±0.13|0.46±0.14|0.50±0.34|0.60±0.15|0.47±0.18|0.53±0.15|
> ||Jerk|2.86±1.90|2.96±1.95|3.49±3.88|3.71±1.73|3.62±2.03|4.51±1.59|
> ||Fuel|0.314±0.026|0.314±0.025|0.354±0.053|0.321±0.019|0.322±0.028|0.327±0.019|
>
> During training, all RL-based controllers improve smoothness and efficiency satisfying safety constraints over the default FVD car-following model setting. The more critical distinction appears in deployment. While several baselines achieve comparable rewards, they suffer from noticeably higher costs and degraded motion quality. In contrast, our full model attains the lowest deployment cost below the budget, together with low oscillation ratio, the lowest jerk, and the lowest fuel consumption.
>
> [1] doi:10.1016/j.trc.2022.103744.
>
> [Limitation] Please see our responses to your "[Question 2]". Beyond autonomous driving, the proposed framework is potentially applicable to a broader range of safety-critical CPS, such as drones and humanoid robots. These applications often require simulator-based training, while the real deployment environment may contain hidden characteristics that are difficult to be modeled explicitly. In such cases, the latent context encoder and risk adaptation mechanism in our framework can support safer and more efficient policy transfer under the Sim2Real gap.

---

### Decision · Program_Chairs · 2026-04-30

**Decision:**

Accept (regular)

**Comment:**

This work proposes a novel framework to mitigate the Sim2Real gap in cyber-physical systems by enabling safe and efficient policy transfer. It leverages probabilistic latent context adaptation within meta-RL to infer the environment's latent representation from simulated data. Crucially, it incorporates a distributional RL formulation that allows the risk level of the deployed policy to be adjusted dynamically at inference time based on the estimation accuracy of the latent context variable.

The reviewers were highly satisfied with the paper's solid theoretical foundation and the well-structured proofs presented. The integration of distributional RL with latent context adaptation offers a powerful and novel approach to managing uncertainty during policy deployment. Given the comprehensive theoretical contributions, the strong empirical performance, and the promising framework for addressing the Sim2Real gap, I recommend acceptance.

For the final version, further discussion could be beneficial on the practical balance between promoting safety early in deployment versus maintaining high efficiency through fast policy adaptation, alongside providing clearer operational guidelines for the dynamic risk adjustment mechanism. There are also some closely related work that study sim2real gap through learning the latent embedding of source and target domains and use this representation mismatch to do data filtering (Cross-Domain Policy Adaptation by Capturing Representation Mismatch) or approximate the target dynamics model (MOBODY: Model-Based Off-Dynamics Offline Reinforcement Learning). It would be helpful for the authors to discuss the connection in the final version.